# RETA: Real-Time and Expressive Talking Head Animation without Emotion Label

## Abstract

Generating photorealistic and expressive talking heads from audio faces a generative trilemma, forcing a trade-off between real-time performance, lip-sync accuracy, and emotional fidelity. We propose RETA, an end-to-end framework that resolves this trilemma. The core of RETA is a novel strategy that disentangles the audio signal into two representations. First, for robust lip-sync, we use a 3DMM as a differentiable bridge, providing strong geometric guidance within an end-to-end model to prevent error accumulation. Second, for nuanced expression, a dynamic emotion embedding is learned from audio in a completely label-free manner; this is achieved by combining cross-modal knowledge distillation from a visual expert with a novel cross-synthesis consistency loss to ensure the representation is identity-agnostic. These representations are then hierarchically injected into a single-pass GAN generator for disentangled control. RETA establishes a new state-of-the-art (SOTA) by outperforming previous methods across all key metrics, while generating high-fidelity video at speeds exceeding 55 FPS. *Code will be available upon publication.*

## 1 Introduction

Audio-driven portrait animation (Jamaludin et al., 2019; Zhou et al., 2019; Pataranutaporn et al., 2021), the synthesis of photorealistic talking head videos from a single static image and a driving audio signal, is a rapidly emerging field. Its wide-ranging applications in virtual avatars, film production, and online education highlight its immense practical value (Wang et al., 2022c; Yu et al., 2023).

A practical SOTA system must satisfy three pillars: (1) **Lip-Synchronization**: This pillar (Prajwal et al., 2020b; Sun et al., 2022; Wang et al., 2023b) represents the fundamental requirement of audio-visual coherence. It demands precise temporal alignment between the synthesized mouth movements and the phonemes of the driving audio. (2) **Expressive Naturalness:** Beyond accurate lip-sync, the animation must exhibit emotional congruence with the audio input (Gururani et al., 2023; Tian et al., 2024). This requires translating cues in the audio—such as prosody, pitch, and cadence—into corresponding and realistic facial expressions (Peng et al., 2023; Kim et al., 2025). Without this emotional fidelity, the resulting animation appears unnatural and fails to convey the presence of a living subject. (3) **Real-Time Performance:** For real-world viability (Guo et al., 2024; Ji et al., 2024; Wang et al., 2025b; Ye et al., 2024), particularly in interactive applications, the system must operate with low latency and minimal computational overhead. Fast inference is therefore essential for smooth real-time performance and broader adoption.

Unfortunately, current methods face a generative trilemma, forced to sacrifice one pillar to achieve others, preventing a holistic solution. As shown in Table 1, we can categorize recent approaches into three groups and identify their inherent compromises. (1) *Multi-Stage Geometric Pipelines*, such as SadTalker (Zhang et al., 2023b), utilize intermediate geometric representations such as 3DMM coefficients. While this approach effectively decouples motion from appearance, it is highly susceptible to error accumulation (Tan et al., 2024a). Inaccuracies introduced in the initial audio-to-geometry prediction stage are propagated and often amplified during final rendering, which degrades visual quality and leads to imprecise lip movements, thereby failing to satisfy Pillar 1. (2) *Label-Dependent Emotional Models*, including EAMM (Ji et al., 2022) and EAT (Gan et al., 2023), rely on external inputs like reference videos or discrete emotion labels to guide expression synthesis. This approach exhibits two primary flaws. First, it imposes an onerous requirement on users to provide

Table 1: Comparison of our RETA with prior SOTA methods. RETA is among the first to simultaneously address the trilemma, offering an end-to-end, real-time pipeline with expressive emotion control without relying on explicit geometric priors or emotion labels. Exp. Nat.: Expressive Naturalness.

| Method | Exp. Nat. | Lip-sync | Real-Time |
|---|---|---|---|
| SadTalker (Zhang et al., 2023b), GMTalker (Xia et al., 2025), Styleheat (Yin et al., 2022), LES-Talker (Feng et al., 2025), RealPortrait (Ye et al., 2024) | ✗ | ✗ | ✓ |
| EMMN (Tan et al., 2023), EAT (Gan et al., 2023), Style2Talker (Tan et al., 2024b), DREAM-Talk (Ma et al., 2023) | ✗ | ✓ | ✗ |
| EMO (Tian et al., 2024), Aniportrait (Wei et al., 2024), AniTalker (Liu et al., 2024), EchoMimic (Chen et al., 2025), KDTalker (Yang et al., 2025) | ✓ | ✓ | ✗ |
| RETA (ours) | ✓ | ✓ | ✓ |

suitable emotional references. Second, by depending on fixed labels, it overlooks the rich, continuous emotional cues inherent in the audio itself, resulting in a limited and often monotonous range of expressions. This fundamentally compromises Pillar 2. (3) *End-to-End Diffusion Models*, such as EMO (Tian et al., 2024), significantly advance photorealistic fidelity and expressive naturalness. However, their iterative sampling process incurs prohibitive computational costs and high inference latency, making them impractical for real-time use and failing Pillar 3. Consequently, a clear gap exists, as **no single framework effectively resolves the generative trilemma.**

To break this impasse, we propose **RETA**, a novel framework for **R**eal-time and **E**xpressive **T**alking head **A**nimation that simultaneously achieves all three pillars within a single, end-to-end architecture. Our key innovations are validated by SOTA results: (1) **Pillar 1 (Lip-Synchronization)**: We introduce a 3DMM as a differentiable bridge within our end-to-end framework. This avoids the error accumulation of multi-stage pipelines, achieving a superior lip-sync confidence score (6.21 $Sync_c$ on HDTF) that approaches ground-truth levels. (2) **Pillar 2 (Expressive Naturalness)**: Our framework operates entirely without emotion labels. We distill nuanced emotional cues from audio via cross-modal knowledge transfer, while a novel cross-synthesis consistency loss ensures the learned emotion is identity-agnostic. This yields photorealism on par with leading diffusion models (14.80 FID) while rendering highly accurate expressions (62.35% $ACC_e$). (3) **Pillar 3 (Real-Time Performance)**: RETA is engineered for efficiency, using a lightweight geometric prior and a single-pass GAN generator. This results in true real-time inference at over 55 FPS—a 2.5-3.6× speedup over strong competitors and orders of magnitude faster than iterative diffusion models. Overall, the motion and emotion representations are integrated via a novel Hierarchical Injection strategy, which introduces the geometric prior to early layers and emotion features to later layers, preventing interference. Fidelity is further enhanced by a Temporal Variance Masking (TVM) module that focuses reconstruction on dynamic facial regions. Through these innovations, RETA provides a unified and empirically-validated solution that resolves the generative trilemma.

In summary, our key contributions are:

- We propose RETA, the first end-to-end framework to resolve the trade-offs of prior work by simultaneously achieving accurate lip-sync, nuanced expressiveness, and real-time performance.
- We introduce a novel use of a 3DMM as a differentiable bridge, which provides strong geometric guidance for robust lip-sync while avoiding the error accumulation common in multi-stage pipelines.
- We present a label-free strategy to learn continuous emotion representations from audio via cross-modal distillation, coupled with a Hierarchical Injection mechanism that strategically integrates motion and emotion signals into different generator layers to prevent interference between the coarse-grained geometric signals (for structure) and the fine-grained emotion signals (for detail).
- Extensive experiments on the MEAD and HDTF benchmarks show RETA outperforms SOTA methods in photorealism, lip-synchronization, and expressiveness, all while operating in real-time.

## 2 RELATED WORKS

**Geometric Prior-based Methods.** A dominant strategy in talking head generation is to employ a two-stage pipeline reliant on geometric priors (Ye et al., 2024). These priors evolve from 2D landmarks (Chen et al., 2019) to more structured 3D Morphable Models (3DMMs) used in methods like SadTalker (Zhang et al., 2023b), GMTalker (Xia et al., 2025), and LES-Talker (Feng et al., 2025). More recently, SyncTalk (Peng et al., 2024) leverages a NeRF-based pipeline with a 3D

facial blendshape model to enhance expression control and pose stability. These methods first predict 3DMM coefficients from audio to drive a separate renderer, a design that suffers from compounding errors. In contrast, RETA's unified framework uses the 3DMM as a differentiable regularizer—not a supervised target—learning an implicit geometric prior via backpropagation from the final video to avoid error accumulation.

**Emotional Control Methods.** Emotional control has been a central challenge, with methods evolving from explicit supervision to more sophisticated, audio-driven techniques. Early approaches, such as the GCN-based method of Sinha et al. (2022), relied on discrete, one-hot emotion vectors, which limited expressiveness to predefined categories. A second line of work achieved more dynamic control by using external guidance, either by transferring style from a reference video, as in EAMM (Ji et al., 2022) and DREAM-Talk (Zhang et al., 2023a), or by using text prompts, as seen in Style2Talker (Tan et al., 2024b) and EAT (Gan et al., 2023). While flexible, these methods all depend on auxiliary, non-audio inputs. A more advanced paradigm aims to derive emotion directly from the rich cues within the audio signal itself, typically by disentangling it from other motion factors. PC-AVS (Zhou et al., 2021), for example, focuses on separating head pose from lip motion. More directly targeting emotion, EVP (Ji et al., 2021) uses unimodal audio cross-reconstruction, GC-AVT Liang et al. (2022) employs input-space masking of facial regions, and PD-FGC (Wang et al., 2023a) utilizes statistical decorrelation. Although these audio-driven strategies are self-contained, they often define emotion indirectly through structural or statistical separation. RETA advances this by introducing a more direct learning signal: without relying on external supervision or discrete labels, our framework autonomously learns to extract nuanced emotional cues from the input audio's acoustic properties. This is achieved via cross-modal distillation from a visual expert, which transfers knowledge without requiring explicit emotion labels. For a detailed comparison of our disentanglement strategy against prior methods, please see Appendix B.

**Diffusion-based Methods.** The advent of diffusion models (Ho et al., 2020) extends to talking head generation (Du et al., 2023; Ma et al., 2023), where two main strategies emerge. One line of work uses diffusion models as renderers to translate geometric priors into photorealistic frames, as seen in AniPortrait (Wei et al., 2024), EchoMimic (Chen et al., 2025). A second approach, demonstrated by AniTalker (Liu et al., 2024) and KDTalker (Yang et al., 2025), uses diffusion as a motion generator to create dynamic keypoint sequences from audio. More recently, EmotiveTalk (Wang et al., 2025a) proposed a video diffusion framework that uses a Vision-guided Audio Information Decoupling (V-AID) approach to generate separate representations for lips and expression. While these techniques achieve SOTA visual quality, their high computational cost and slow inference speeds make them unsuitable for real-time applications. In contrast, RETA's single-pass GAN architecture is designed for efficiency, enabling real-time performance.

## 3 METHOD

As illustrated in Fig. 1, our proposed framework, RETA, synthesizes a photorealistic talking-head sequence $\hat{\mathbf{I}}_{1:T}$ from a single identity image $\boldsymbol{I}^i$ and a corresponding audio clip $\mathbf{A}_{1:T}$. Our methodology is built on the principle of strategically orchestrating the knowledge encapsulated in powerful, pre-trained models to solve higher-level challenges. This allows us to focus our contributions on the primary bottleneck: the generation of novel, disentangled control signals and their effective integration. Specifically, we leverage a pre-trained Wav2Vec 2.0 $\mathcal{E}_A$ for audio encoding, a ViT as a visual "teacher" $\mathcal{T}$, and SyncNet for lip-sync supervision, while adapting the EDTalk generator ($\mathcal{G}$) and image encoder ($\mathcal{E}_I$) as a high-fidelity synthesis backbone. The core of our innovation lies in how we generate and integrate two novel audio-driven signals—a geometric prior $\boldsymbol{f}_M$ and an emotion embedding $\boldsymbol{f}_E$—to control this backbone. The overall generation process consists of two main stages:

- **Complementary Feature Learning** (Fig. 1(a)): An audio encoder $\mathcal{E}_A$ first extracts features $\boldsymbol{f}_A = \mathcal{E}_A(\mathbf{A}_{1:T})$ from the input audio. These features are then fed into two parallel branches: a **Geometric Prior Learning** branch, which predicts an implicit motion prior $\boldsymbol{f}_M$, and an **Emotion Representation Learning** branch, which extracts a dynamic emotion embedding $\boldsymbol{f}_E$. While speech geometry and emotional expression are often correlated, our framework is designed to functionally decouple these aspects into two complementary streams. This approach encourages each branch to specialize: the geometric branch focuses on predictable, speech-driven motion for robust lip-sync, while the emotion branch models the nuanced expressions tied to audio

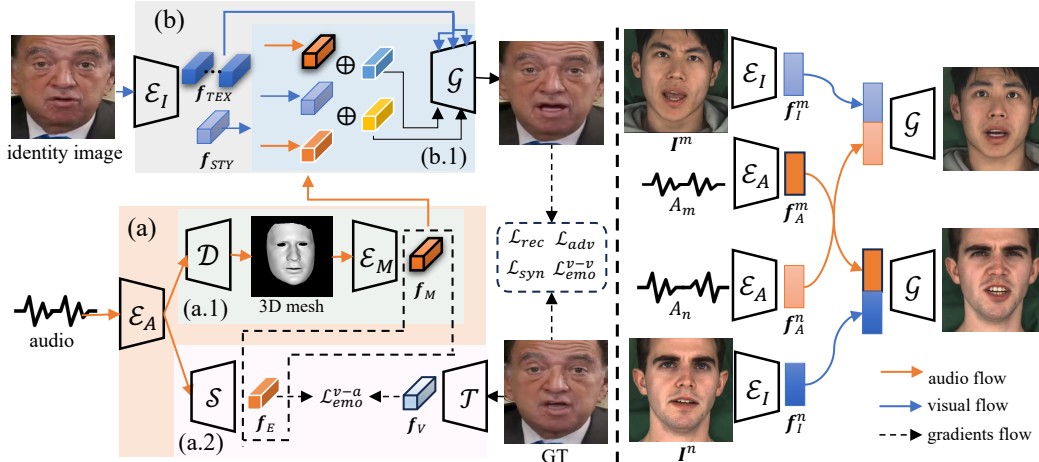

Figure 1: RETA overview: The main pipeline (left) generates a talking head video from a single image and audio. It first performs Disentangled Feature Learning to extract a geometric prior (a.1) and an emotion representation (a.2). These features are then integrated into the Hierarchical Face Generation stage (b) using a core Hierarchical Injection mechanism (see Appendix Fig. 4 for details). The emotion representation is learned without labels via a cross-synthesis strategy (right). Components: $\mathcal{E}_I$ (Image Encoder), $\mathcal{E}_A$ (Audio Encoder), $\mathcal{E}_M$ (Motion Encoder), $\mathcal{D}$ (Decoder), $\mathcal{S}$ (Student), $\mathcal{T}$ (Teacher), and $\mathcal{G}$ (Generator).

prosody. This functional separation is crucial for resolving the ambiguity in audio-to-video synthesis by separating predictable, speech-correlated geometric motion from more stochastic, nuanced emotional expressions.

- **Hierarchical Face Generation** (Fig. 1(b)): The synthesis process is orchestrated by a GAN-based generator $\mathcal{G}$, which hierarchically integrates the learned features. The process begins by encoding the source identity image $\boldsymbol{I}^i$ using a vision encoder $\mathcal{E}_I$. This yields two distinct representations: a global, non-spatial style code $\boldsymbol{f}_{STY}$ encapsulating the holistic appearance, and a pyramid of multi-scale spatial features $\boldsymbol{f}_{TEX}$ that retain rich, high-fidelity texture details. The generator is first conditioned on $\boldsymbol{f}_{STY}$ to set the subject's foundational appearance. Adopting a coarse-to-fine approach, the geometric prior $\boldsymbol{f}_M$ is injected into the early layers of $\mathcal{G}$ to dictate the overall facial structure and head motion. In parallel, the emotion embedding $\boldsymbol{f}_E$ modulates the later, fine-grained layers to synthesize nuanced expressions. To guarantee robust identity preservation, the identity feature $\boldsymbol{f}_{TEX} = \mathcal{E}_I(\boldsymbol{I}^i)$ is also fed into the generator along with these control signals.

The framework is trained end-to-end using a self-reconstruction objective, where a composite loss supervises generated video $\hat{\mathsf{I}}_t$ against the ground-truth video $\mathsf{I}_t^{\text{GT}}$. A key part of our training is the *Emotion Disentanglement Strategy* (Fig. 1, Right), which uses a cross-synthesis consistency loss to learn the emotion embedding $\boldsymbol{f}_E$ without any labels. The following sections detail these components. For ease of exposition, all notations used in this paper are summarized in Table 5.

## 3.1 FUNCTIONALLY DISENTANGLED FEATURE LEARNING

As highlighted by Zhou et al. (2020), a primary challenge in talking head generation is the ambiguity of direct audio-to-pixel mapping, which can lead to inaccurate lip movements and unnatural motion. This arises from the inherent one-to-many relationship, as a single phoneme can correspond to diverse articulatory and emotional facial movements. To address this, our framework approaches the problem through functional disentanglement, mapping the audio signal to two complementary and more predictable streams: geometric motion and emotion representation. Though these two aspects may be correlated in the input audio (e.g., a smile affects both geometry and emotion), our architecture and targeted loss functions encourage each branch to specialize in a distinct function. Specifically, we employ the pre-trained Wav2vec 2.0 model (Baevski et al., 2020) as our audio encoder, $\mathcal{E}_A$, to extract rich features, $\boldsymbol{f}_A = \mathcal{E}_A(\mathbf{A}_{1:T})$, from the raw audio. These features then serve as the common input for the two parallel branches described in the following sections.

**Geometric Prior Learning.** We model the geometric motion stream using a 3D Morphable Model (3DMM) (see Appendix C.1), which transforms the ill-posed audio-to-pixel problem into a constrained task of predicting low-dimensional facial parameters. This parametric representation inherently disentangles the strong audio-lip correlation from weaker correlations with head pose or blinks, mitigating the one-to-many ambiguity (Xia et al., 2025).

Crucially, we are among the first to use the 3DMM not as a pre-computed target for supervision, but as a differentiable geometric regularizer within an end-to-end framework. This avoids the error accumulation common in previous multi-stage methods (Zhang et al., 2023b; Feng et al., 2025), where early prediction errors degrade the final rendering. Moreover, our "differentiable bridge" design offers two key advantages. First, it constrains the optimization: instead of mapping audio to an unconstrained, high-dimensional pixel space, the model predicts a few low-dimensional 3DMM parameters, shrinking the search space, providing strong geometric guidance, and steering optimization toward a reasonable, anatomically plausible optimum—thereby addressing the one-to-many ambiguity by focusing on predictable, speech-driven facial motion (Pillar 1). Second, it improves inference speed (Pillar 3), as predicting a small set of parameters requires only a lightweight network, unlike estimating computationally expensive dense motion fields.

As illustrated in Fig. 1(a.1), audio features $\boldsymbol{f}_A$ given by $\mathcal{E}_A$ are fed into a motion decoder $\mathcal{D}$ (a Temporal Convolutional Network, TCN), which predicts a sequence of 3DMM expression parameters $\boldsymbol{p}_t^{\exp}$. These parameters, in turn, define a sequence of 3D facial meshes $\boldsymbol{S}_{1:T} = \{\boldsymbol{s}_1, ..., \boldsymbol{s}_T\}$ via the standard 3DMM formulation $\boldsymbol{s}_t = \bar{\boldsymbol{s}} + \boldsymbol{U}^{\mathrm{id}}\boldsymbol{p}^{\mathrm{id}} + \boldsymbol{U}^{\exp}\boldsymbol{p}_t^{\exp}$, where $\bar{\boldsymbol{s}}$ is the mean face shape, and $\boldsymbol{U}^{\mathrm{id/exp}}$ are the 3DMM bases. A key aspect of our design is the strategic handling of identity and expression parameters within the 3DMM. We make the following choices:

- Fixed Identity Prior: We do not learn the identity parameter $\boldsymbol{p}^{\mathrm{id}}$. Instead, following Ren et al. (2021), we fix it to a prior value representing a generic mean face. The rationale is that the low-dimensional $\boldsymbol{p}^{\mathrm{id}}$ only captures coarse, low-frequency aspects of facial identity (e.g., bone structure) and cannot preserve the high-frequency details (e.g., skin texture, eye color, and fine wrinkles). We therefore delegate this task to the vision encoder $\mathcal{E}_I$, which extracts rich, high-dimensional appearance features ($\boldsymbol{f}_{STY}, \boldsymbol{f}_{TEX}$) directly from the source image, which will be discussed in Subsection 3.2.

- Expression for Geometry: While $\boldsymbol{p}_t^{\exp}$ is learned, we use it primarily to guide speech-related geometry (e.g., lip and jaw movements). The 3DMM's expression space is too coarse for subtle emotions, so we use a separate, dedicated emotion feature $\boldsymbol{f}_E$ to synthesize fine-grained expressions.

The resulting mesh sequence $\boldsymbol{S}_{1:T}$ is then encoded by a lightweight CNN-based motion encoder $\mathcal{E}_M$ to produce the final motion feature $\boldsymbol{f}_M$. Notably, neither $\boldsymbol{p}_t^{\exp}$ nor the intermediate meshes are directly supervised; they serve as a differentiable geometric bridge from audio to synthesis. Supervision comes only from the rendered output, letting $\mathcal{D}$ learn an optimal geometric prior via end-to-end backpropagation so the geometry is optimized for final image generation rather than for matching a potentially noisy intermediate representation.

**Emotion Representation Learning.** As shown in Fig. 1(a.2), to learn a rich, visually-grounded emotion representation from audio without labels, we propose a cross-modal knowledge distillation strategy, as inspired by Jin et al. (2023); Sun et al. (2024). The approach employs a visual "teacher" $\mathcal{T}$, which is a Vision Transformer (ViT) pre-trained on a large-scale facial expression dataset and acts as an expert emotion extractor, producing an embedding $\boldsymbol{f}_V = \mathcal{T}(\mathbf{I}^{GT})$ from each ground-truth video frame. Concurrently, our trainable "student" $\mathcal{S}$ (a TCN-based emotion decoder) processes audio features to generate a corresponding embedding $\boldsymbol{f}_E = \mathcal{S}(\boldsymbol{f}_A)$. The student is trained to mimic the teacher by minimizing the cosine distance between their embeddings: $\mathcal{L}_{emo}^{v-a} = 1 - \cos(\boldsymbol{f}_E, \mathrm{sg}(\boldsymbol{f}_V))$, where $\cos(\cdot, \cdot)$ is the cosine similarity and $\mathrm{sg}(\cdot)$ is the stop-gradient operator. This process effectively transfers nuanced emotional understanding from the visual domain to our audio network and bypasses the need for manual annotations.

## 3.2 Hierarchical Face Generation

The final synthesis stage targets both high-fidelity rendering and real-time performance (all Pillars) using a conditional GAN generator $\mathcal{G}$, adapted from warping-based models LIA (Wang et al., 2022b) and EDTalk (Tan et al., 2024a). We initialize $\mathcal{G}$ and its encoder $\mathcal{E}_I$ with EDTalk's pretrained weights to leverage robust facial-animation priors. Rather than synthesizing pixels from scratch, the generator

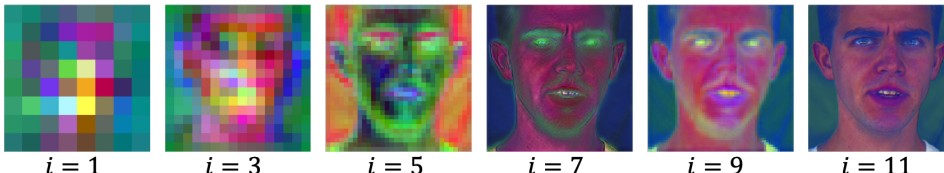

$$i = 1 \qquad i = 3 \qquad i = 5 \qquad i = 7 \qquad i = 9 \qquad i = 11$$

Figure 2: Visualization of the coarse-to-fine synthesis process within the generator. Each image displays the output from a different generator layer, where i denotes the output of the i-th layer.

predicts a dense optical flow field and an occlusion mask to warp source-image features into the animated frame. Architectural details are in Appendices C.2 and C.4.

The core challenge here is to effectively integrate the diverse control signals—identity, motion, and emotion—into warping without interference. To this end, we first encode the source identity image $\boldsymbol{I}^i$ with the encoder $\mathcal{E}_I$ into two complementary representations. A non-spatial **global style code**, $\boldsymbol{f}_{STY}$, is extracted from the final layer of $\mathcal{E}_I$ to encapsulate the holistic appearance identity. A pyramid of multi-scale **spatial texture features**, $\boldsymbol{f}_{TEX}$, is extracted from $\mathcal{E}_I$'s intermediate layers. These features retain the rich, spatially-aligned textures and structural details that serve as the high-fidelity "content canvas" to be animated. Together, $\boldsymbol{f}_{STY}$ and $\boldsymbol{f}_{TEX}$ ensure that the subject's identity remains stable and detailed across the entire generated sequence. The generator $\mathcal{G}$ then renders the target frame $\hat{\mathbf{I}}_t$ by synergizing them with the audio-driven motion prior $\boldsymbol{f}_M$ and the emotion embedding $\boldsymbol{f}_E$.

**Analysis of the Generation Process.** As shown in Fig. 2, we analyze the generator's synthesis process using Principal Component Analysis (PCA) on its intermediate features. By projecting features from different layers onto their top principal components and visualizing them as RGB channels, we observe a distinct coarse-to-fine progression. Early-stage features establish global structure and layout, while later-stage features progressively add intricate local details and textures. This finding motivates our Hierarchical Injection strategy.

**Hierarchical Injection.** Informed by our analysis, we introduce a Hierarchical Injection strategy to align our control signals with the generator's synthesis stages. This design maximizes controllability by separating the "what to animate" content from the "how to animate" instructions. The "how to animate" instructions are delivered through a composite control signal, $\boldsymbol{f}_{CTRL}$, which modulates the generator's behavior at different layers. For the early, coarse-grained layers that control global structure, the control signal $\boldsymbol{f}_{CTRL} = \boldsymbol{f}_{STY} + \boldsymbol{f}_M$, ensuring robust structural animation and precise lip-sync (Pillar 1). In contrast, for the latter, fine-grained layers which refine local details, $\boldsymbol{f}_{CTRL} = \boldsymbol{f}_{STY} + \boldsymbol{f}_E$. This allows for the synthesis of nuanced, high-frequency emotional details (Pillar 2) without disrupting the underlying facial geometry. The hierarchical control is implemented via Adaptive Instance Normalization (AdaIN) (Huang & Belongie, 2017). Concretely, $\boldsymbol{f}_{CTRL}$ is first mapped through a learned affine transformation AF to produce style parameters $(\alpha, \beta)$, which then stylize the generator's intermediate feature map $\boldsymbol{f}_{INT}^j$ for a specific layer $j$:

$$\alpha, \beta = \text{AF}(\boldsymbol{f}_{CTRL}), \qquad \text{AdaIN} = \alpha \left( \frac{\boldsymbol{f}_{INT}^j - \mu(\boldsymbol{f}_{INT}^j)}{\sigma(\boldsymbol{f}_{INT}^j)} \right) + \beta, \tag{1}$$

where $\mu(\cdot)$ and $\sigma(\cdot)$ denote the channel-wise mean and standard deviation.

This creates two synergistic pathways within the generator. The control pathway uses the AdaIN layers to translate the control codes $(\boldsymbol{f}_{STY}, \boldsymbol{f}_M, \boldsymbol{f}_E)$ into the precise instructions needed to generate the optical flow field. The content pathway, in parallel, supplies the spatial texture features $\boldsymbol{f}_{TEX}$ via skip connections. The generator then uses the predicted flow field to warp the detailed "content" from $\boldsymbol{f}_{TEX}$, effectively animating the source identity according to the audio-driven commands.

We note that the high-level principle of staged generation is also explored in diffusion models (Peng et al., 2025; Liang et al., 2025), which may use early denoising steps for motion and later steps for detail. Our Hierarchical Injection is architecturally distinct: it operates across the spatial layers of a single-pass GAN to resolve interference between explicitly disentangled motion ($\boldsymbol{f}_M$) and emotion ($\boldsymbol{f}_E$) signals. This structural choice is crucial for achieving real-time performance and, as confirmed by our ablation study (Table 4), is essential for preventing the feature clash that would otherwise degrade both lip-sync and expression.

### 3.3 Self-reconstruction Training Strategy

We train the framework end-to-end using a self-reconstruction strategy, where the generated video is supervised against the ground-truth. The composite loss is:

$$\mathcal{L} = \mathcal{L}_{rec} + \lambda_{adv}\mathcal{L}_{adv} + \lambda_{syn}\mathcal{L}_{syn} + \lambda_{emo}(\mathcal{L}_{emo}^{v-v} + \mathcal{L}_{emo}^{v-a}) \tag{2}$$

These terms correspond to a reconstruction loss, an adversarial loss, a lip-sync loss, and an emotion perceptual loss, with weights set following prior work (Tan et al., 2024a). Since the loss terms $\mathcal{L}_{emo}^{v-v}$ and $\mathcal{L}_{emo}^{v-a}$ share the same formulation and objective, they are assigned an equal weight. In addition, we introduce an Emotion Disentanglement Strategy that decouples emotion by applying losses across different videos. Below we elaborate on each component.

**Reconstruction Loss.** Following prior work (Gan et al., 2023; Tan et al., 2024a), we begin with a composite reconstruction loss, $\mathcal{L}_{rec}$, that combines a standard L1 loss for pixel-level accuracy with a VGG-19 based perceptual loss (Simonyan & Zisserman, 2014) to preserve high-frequency details. This baseline loss is formulated as: $\mathcal{L}_{rec} = \lambda_{L1}||\hat{y} - y||_1 + \lambda_{perc}||\phi(\hat{y}) - \phi(y)||_1$. However, this objective treats all pixels equally. To specifically enhance the reconstruction quality of dynamic regions, we augment this with an additional masked L1 term, guided by a dynamically generated motion mask $\mathbf{M}_{motion}$. This mask is produced by our proposed **Temporal Variance Masking (TVM)**, a lightweight, parameter-free algorithm. TVM operates on the principle that spatial regions exhibiting high variance along the temporal dimension correspond to dynamic content (e.g., mouth movements), whereas low-variance regions represent the static background. The mask is thus constructed by isolating these high-variance areas (see Appendix for details). Our final reconstruction loss integrates this dynamic component, adaptively concentrating the model's capacity on these challenging moving regions:

$$\mathcal{L}_{rec} = \lambda_{L1}||\hat{y} - y||_1 + \lambda_{perc}||\phi(\hat{y}) - \phi(y)||_1 + \lambda_{motion}||(\hat{y} - y) \odot \mathbf{M}_{motion}||_1 \tag{3}$$

where $\lambda_{motion}$ weights the motion-focused reconstruction term.

**Adversarial Loss.** To enhance realism (Zhou et al., 2021; Zhang et al., 2023b), we employ an adversarial loss. Our generator $\mathcal{G}$ is trained to minimize a relativistic average objective, while the discriminator $\mathcal{D}_{adv}$ is trained on a symmetric loss:

$$\mathcal{L}_{adv} = \mathbb{E}_{\hat{y},y}\left[\text{softplus}(\mathcal{D}_{adv}(y) - \mathcal{D}_{adv}(\hat{y}))\right] \tag{4}$$

where $\hat{y}$ and $y$ are generated and real samples, respectively.

**Syncnet Loss.** To ensure precise temporal alignment between the generated lip movements and the input audio, we incorporate a lip-sync loss $\mathcal{L}_{sync}$. We leverage a pre-trained SyncNet (Chung & Zisserman, 2016a) model, a widely-used expert network for audio-visual synchronization (Gan et al., 2023). The loss is formulated to maximize the cosine similarity between these two embeddings:

$$\mathcal{L}_{sync} = 1 - \cos(\boldsymbol{v}, \boldsymbol{s}) = 1 - \frac{\boldsymbol{v} \cdot \boldsymbol{s}}{\max(||\boldsymbol{v}||_2||\boldsymbol{s}||_2, \epsilon)} \tag{5}$$

where $\boldsymbol{v}$ and $\boldsymbol{s}$ are the embeddings extracted by the SyncNet's visual and audio encoders, respectively. The term $\epsilon$ is a small constant added for numerical stability.

**Emotion Perceptual Loss.** To ensure the emotional fidelity of the final rendered images, we employ a perceptual loss that operates in the feature space of the pre-trained ViT model. Repurposing the ViT teacher model $\mathcal{T}$ as a fixed perceptual supervisor, we extract emotion logits from the generated image ($z_{\text{gen}}$) and its ground-truth (GT) frame ($z_{\text{gt}}$). The perceptual emotion loss, $\mathcal{L}_{emo}^{v-v}$, is then defined as the KL divergence (Goodfellow et al., 2016) between their Softmax probability distributions:

$$\mathcal{L}_{emo}^{v-v} = D_{\text{KL}}\left(\mathcal{P}(\text{sg}(z_{\text{gt}})) \parallel \mathcal{P}(z_{\text{gen}})\right) \tag{6}$$

where the stop-gradient $sg(\cdot)$ on $z_{\text{gt}}$ emphasizes that the GT distribution is a fixed target. Minimizing $\mathcal{L}_{emo}^{v-v}$ guides the generator to produce images that are perceptually aligned with the true emotion.

**Emotion Disentanglement Strategy.** To ensure the learned emotion embedding $\boldsymbol{f}_E$ is truly disentangled from identity-specific attributes, we introduce a cross-synthesis consistency loss. This strategy enables self-supervision by forcing the model to recombine identity and emotion features from different source videos. For any two training samples, $(\mathbf{I}^m, \boldsymbol{A}_m)$ and $(\mathbf{I}^n, \boldsymbol{A}_n)$, we extract their respective identity features ($\boldsymbol{f}_I^m, \boldsymbol{f}_I^n$) and audio-derived emotion embeddings ($\boldsymbol{f}_E^m, \boldsymbol{f}_E^n$). We then perform a feature swap to generate a cross-synthesized image, $\hat{\mathbf{I}}^{mn} = \mathcal{G}(\boldsymbol{f}_I^m, f_E^n)$., by feeding the generator the identity of sample $m$ but the emotion of sample $n$. If disentanglement is successful, $\hat{\mathbf{I}}^{mn}$ should retain the identity of $\mathbf{I}^m$ but exhibit the emotional expression of $\mathbf{I}_n$. Consequently, the

Table 2: Quantitative comparisons with SOTA methods. ↑ denotes higher the better, ↓ denotes lower the better. The best result is in **bold**, and the second-best is underlined. * denotes diffusion methods.

| Method | Inference | MEAD | | | | | HDTF | | | |
|---|---|---|---|---|---|---|---|---|---|---|
| | | PSNR ↑ | SSIM ↑ | FID ↓ | Sync$_c$ ↑ | ACC$_e$ ↑ | PSNR ↑ | SSIM ↑ | FID ↓ | Sync$_c$ ↑ |
| EAMM | 16.87 | 17.21 | 0.60 | 33.27 | 1.98 | 32.90 | 19.17 | 0.62 | 40.36 | 2.33 |
| EAT | 12.23 | 21.33 | 0.70 | 20.48 | 5.17 | **69.35** | 22.15 | 0.70 | 27.58 | 4.32 |
| SadTalker | 21.74 | 21.39 | 0.68 | 25.84 | 5.36 | 29.33 | 21.20 | 0.60 | 15.33 | 5.66 |
| RealPortrait | 3.77 | 20.55 | 0.62 | 21.22 | 5.67 | 43.55 | 21.41 | 0.69 | 27.27 | 5.32 |
| AniTalker* | 3.57 | 20.90 | 0.70 | 24.17 | 4.53 | 33.49 | 21.82 | 0.68 | 18.01 | 4.69 |
| Aniportrait* | 3.84 | 21.23 | 0.69 | 23.17 | 2.35 | 48.22 | 21.69 | 0.61 | 15.02 | 3.18 |
| EchoMimic* | 0.23 | 21.89 | 0.75 | 19.11 | 4.24 | 39.21 | 22.33 | 0.73 | 17.45 | 4.48 |
| KDTalker* | 15.34 | **22.32** | **0.79** | 16.59 | 4.22 | 42.40 | **23.07** | **0.78** | 16.52 | 4.93 |
| RETA (ours) | **55.32** | 22.19 | 0.77 | 17.09 | **5.84** | 62.35 | 22.82 | 0.75 | **14.80** | **6.21** |
| GT | - | - | - | - | 6.35 | 77.37 | - | - | - | **6.49** |

emotional ground truth for $\hat{\mathbf{I}}^{mn}$ is the original frame $\mathbf{I}^n$. We enforce this consistency by applying $\mathcal{L}_{emo}^{v-v}$ in Eq. (6) between the cross-synthesized image $\hat{\mathbf{I}}^{mn}$ and the emotion source's ground truth $\mathbf{I}^n$.

# 4 EXPERIMENTAL RESULTS

We train and evaluate our model on the MEAD (Wang et al., 2020) and HDTF (Zhang et al., 2021) datasets, covering both controlled and in-the-wild scenarios with video frames resized to 256×256 (Danecek et al., 2022). We benchmark RETA against SOTA methods categorized by their core strategy: emotion label-dependent (EAMM (Ji et al., 2022), EAT (Gan et al., 2023)), geometric prior-based (SadTalker (Zhang et al., 2023b), RealPortrait (Ye et al., 2024)), and diffusion-based (Aniportrait (Wei et al., 2024), AniTalker (Liu et al., 2024), EchoMimic (Chen et al., 2025), KDTalker (Yang et al., 2025)). To rigorously evaluate performance against our three pillars, we employ a holistic suite of metrics: for **Image Quality and Realism**, we use PSNR, SSIM, and the Fréchet Inception Distance (**FID**) (Heusel et al., 2017); for **Lip-Synchronization (Pillar 1)**, we report the SyncNet confidence score (**Sync$_c$**) (Chung & Zisserman, 2016b); for **Expressive Naturalness (Pillar 2)**, we measure emotion accuracy (**ACC$_e$**) (Meng et al., 2019); and for **Real-Time Performance (Pillar 3)**, we report Frames Per Second (**FPS**). More details regarding the datasets, the baselines, the evaluation metrics, and the implementation details are provided in Appendix D.

## 4.1 MAIN RESULTS

As shown in Table 2, RETA establishes a new SOTA by delivering superior performance across all three pillars of talking head generation: lip-synchronization, expressive naturalness, and real-time inference. It successfully resolves the generative trilemma that limits prior work.

**Pillar 1: Superior Lip-Synchronization.** RETA achieves the highest Sync$_c$ score on both datasets (e.g., 6.21 on HDTF), surpassing all baselines and approaching the ground-truth level (6.49). This validates the efficacy of our end-to-end framework, where the 3DMM acts as a differentiable bridge. Unlike multi-stage methods like SadTalker, our design avoids the accumulation of geometric prediction errors, resulting in more precise lip articulation. Furthermore, while diffusion models like KDTalker and AniTalker show strong FID scores, their lower Sync$_c$ scores highlight a key weakness: their tendency to average results blurs fine-grained lip motion, degrading audio-visual coherence. RETA's GAN-based architecture, by contrast, preserves these critical dynamic details.

Table 3: User study results of our RETA with prior SOTA methods.

| Metric | Lip-sync | Realness | Naturalness |
|---|---|---|---|
| RealPortrait | 0.53 | 0.62 | 0.55 |
| Aniportrait | 0.35 | 0.33 | 0.32 |
| KDTalker | 0.49 | 0.65 | 0.65 |
| Ours | 0.66 | 0.71 | 0.75 |

**Pillar 2: Expressive Naturalness without Labels.** RETA excels in generating both photorealistic and emotionally congruent animations. Regarding **photorealism**, our model achieves a top-tier FID score (14.80 on HDTF), proving it can generate images with quality and diversity comparable to leading diffusion models, but without their computational overhead. While some methods like KDTalker report slightly higher PSNR/SSIM values, this points to the classic perception-distortion trade-off (Zhang et al., 2018). These methods over-optimize for pixel-wise similarity, often producing

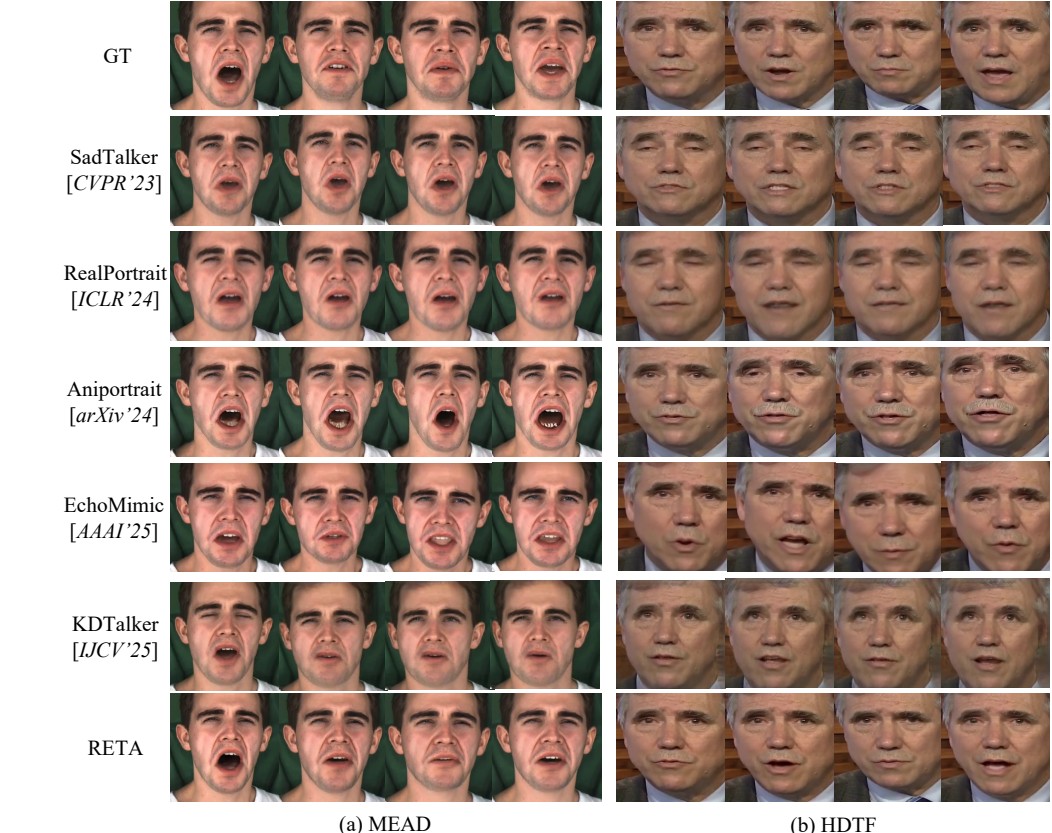

(a) MEAD                    (b) HDTF

Figure 3: Qualitative comparisons. See Appendix E for more visualizations.

overly smooth, "averaged" images that lack realism, as seen qualitatively in Fig. 3. RETA, by contrast, prioritizes perceptual quality, which is confirmed by its strong FID and user study results. On the other hand, in terms of **expressiveness**, RETA achieves a high $ACC_e$ score (62.35% on MEAD), demonstrating its ability to render accurate emotional expressions. Crucially, it does so in a fully label-free manner, a significant practical advantage over methods like EAT, which requires explicit emotion labels to reach its slightly higher score (69.35%). This result validates our cross-modal distillation strategy for learning nuanced emotion directly from audio.

**Pillar 3: Real-Time Performance.** Our framework is designed for efficiency, achieving an inference speed of 55.32 FPS on a single A100 GPU. This is a 2.5× to 3.6× speedup over strong competitors like SadTalker (21.74 FPS) and KDTalker (15.34 FPS), and orders of magnitude faster than diffusion models. This performance, which meets real-time requirements, stems from our single-pass GAN generator and the lightweight geometric prior, avoiding the high latency of iterative sampling in diffusion models.

**Qualitative and Human Evaluation.** The quantitative results are strongly supported by human perception. The qualitative comparisons in Fig. 3 offer a clear visual narrative that corroborates our metric-based findings. While diffusion-based methods like KDTalker and Aniportrait produce smooth outputs, they often fail to capture the precise emotional nuances of the ground truth, resulting in a generalized or muted expression. In contrast, Fig. 3 demonstrates that RETA excels at replicating the specifics of the ground-truth expression and emotion. This expressive fidelity, combined with visibly more accurate lip shapes, directly showcases our model's strengths in both Expressive Naturalness (Pillar 2) and Lip-Synchronization (Pillar 1). Furthermore, a user study

Table 4: Ablation study for main components in the proposed method. Here, GPL: Geometric Prior Learning; ERL: Emotion Representation Learning; HI: Hierarchical Injection; EDS: Emotion Disentanglement Strategy.

| Index | Method | PSNR | SSIM | $Sync_c$ | $ACC_e$ |
|---|---|---|---|---|---|
| A.1 | w/o GPL | 21.21 | 0.68 | 1.85 | 61.72 |
| A.2 | w/o ERL | 21.37 | 0.70 | 5.65 | 41.23 |
| A.3 | w/o 3DMM | 21.05 | 0.66 | 3.23 | 57.41 |
| B.1 | w/o HI | 21.47 | 0.71 | 4.91 | 52.64 |
| B.2 | w/o $f_{STY}$ | 21.63 | 0.70 | 5.75 | 60.15 |
| C.1 | w/o EDS | 21.85 | 0.73 | 5.82 | 45.79 |
| C.2 | w/o TVM | 21.93 | 0.75 | 5.77 | 62.10 |
| - | Full Model | 22.19 | 0.77 | 5.84 | 62.35 |

(Table 3) confirms RETA's superiority, as participants rated it highest in lip-synchronization, photo-realism, and overall naturalness. This convergence of metric-based and human-centric evaluation confirms RETA as a robust and practical solution.

## 4.2 ABLATION STUDY

We summarize the results of the ablation study in Table 4, with more details in Appendix H.

**Disentangled Feature Learning.** Experiments A.1 and A.2 confirm the efficacy of our feature disentanglement. Removing the Geometric Prior Learning (GPL) branch (A.1) cripples lip-sync performance, causing the $Sync_c$ score to plummet from 5.84 to a mere 1.85. Conversely, removing the Emotion Representation Learning (ERL) branch (A.2) devastates emotional expressiveness, with $ACC_e$ dropping from 62.35% to 41.23%. This orthogonal impact provides clear proof that we have successfully isolated the drivers for Pillar 1 (Lip-Sync) and Pillar 2 (Expressiveness) into independent, non-interfering pathways. Furthermore, replacing our 3DMM "differentiable bridge" with an unstructured decoder (A.3) also severely degrades $Sync_c$ (to 3.23), confirming that structured, anatomical guidance is critical for high-fidelity lip motion.

**Hierarchical Generation and Identity Preservation.** Our analysis validates the principled design of the generator. Disabling Hierarchical Injection (HI) and naively concatenating all control signals (B.1) leads to a "feature clash" that harms both lip-sync ($Sync_c$ 4.91) and expression ($ACC_e$ 52.64). This confirms our coarse-to-fine injection strategy is crucial for preventing interference between motion and emotion signals. Similarly, removing the global style anchor $f_{STY}$ (B.2) degrades performance by removing the stable identity context needed for high-fidelity rendering. The spatial texture features, $f_{TEX}$, are a structurally integral component of the warping-based synthesis, and their removal impacts the training stability and leads to a runtime error.

**Emotion and Detail Enhancement.** We isolate the impact of our specialized enhancement modules. Disabling the Emotion Disentanglement Strategy (EDS) (C.1) causes a major drop in $ACC_e$ to 45.79%, proving that our cross-synthesis consistency loss is essential for learning a truly identity-agnostic emotion embedding, free from the "pollution" of identity-specific attributes. Concurrently, removing the Temporal Variance Masking (TVM) module (C.2) results in a noticeable dip in PSNR and SSIM, demonstrating its role in focusing the model's capacity on dynamic regions to enhance fine-grained reconstruction quality.

**Summary of Ablations.** In conclusion, our ablation study confirms RETA's performance relies on the synergistic integration of its components. The GPL and ERL branches are unequivocally the most critical, with their removal causing catastrophic drops in lip-sync (Syncc: 5.84 to 1.85) and expression (ACCe: 62.35% to 41.23%). Other modules like Hierarchical Injection (HI) and the Emotion Disentanglement Strategy (EDS) are also proven vital for preventing feature interference and ensuring identity-agnostic learning. This quantitative breakdown confirms every component is indispensable, working in concert to resolve the generative trilemma.

**Loss Function and Sensitivity Analysis.** Our training objective uses a composite loss to balance lip-synchronization, reconstruction quality, and emotional expressiveness. As detailed in Appendix H, our analysis confirms that each component is critical: removing the lip-sync loss ($\mathcal{L}_{syn}$) severely degrades $Sync_c$, while removing the emotion perceptual losses ($\mathcal{L}_{emo}$) significantly harms $ACC_e$. Furthermore, sensitivity analysis identifies an optimal set of weights that best balances pixel-level reconstruction (PSNR/SSIM) with high-level perceptual quality ($ACC_e$), validating our final configuration.

## 5 CONCLUSION

We propose RETA, an end-to-end framework that resolves the generative trilemma in talking head animation. By integrating a differentiable 3DMM bridge for robust lip-sync and a novel label-free strategy for audio-driven emotion, RETA achieves SOTA performance in lip-synchronization, expressiveness, and photorealism. Its single-pass GAN architecture ensures real-time inference, making it a unified and practical solution that breaks the key trade-offs limiting prior work.

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

# A    MAIN NOTATION INTRODUCTION

Table 5: Meanings of Notations

| Notation | Type | Meaning |
|---|---|---|
| $\boldsymbol{I}^i$ | $B \times 3 \times H \times W$ | The identity image |
| $\mathbf{A}_{1:T}$ | $B \times T \times N$ | The audio clip |
| $\mathcal{E}_I$ | Module | The image encoder |
| $\mathcal{E}_A$ | Module | The audio encoder |
| $\mathcal{E}_M$ | Module | The motion encoder |
| $\mathcal{D}$ | Module | The motion decoder |
| $\mathcal{S}$ | Module | The student module |
| $\mathcal{T}$ | Module | The teacher module |
| $\mathcal{G}$ | Module | The generator |
| $\boldsymbol{f}_E$ | $B \times 512$ | The feature of emotion |
| $\boldsymbol{f}_M$ | $B \times 512$ | The feature of facial motion |
| $\boldsymbol{f}_{TEX}$ | $B \times 512$ | The spatial texture feature |
| $\boldsymbol{f}_{STY}$ | $B \times 512$ | The global style feature |
| $\boldsymbol{f}_{CTRL}$ | $B \times 512$ | The feature of composite control |
| $\boldsymbol{f}_{INT}$ | $B \times C \times H \times W$ | The intermediate feature of generator |
| $\mathcal{L}_{rec}$ | Loss | The reconstruction loss function |
| $\mathcal{L}_{L1}$ | Loss | The L1 loss function |
| $\mathcal{L}_{perc}$ | Loss | The perceptual loss function |
| $\mathcal{L}_{adv}$ | Loss | The adversarial loss function |
| $\mathcal{L}_{syn}$ | Loss | The syncnet loss function |
| $\mathcal{L}_{emo}^{v-v}$ | Loss | The visual-visual emotion perceptual loss function |
| $\mathcal{L}_{emo}^{v-a}$ | Loss | The visual-audio emotion perceptual loss function |
| $\lambda_i$ | Constant | The loss weights |

# B    RELATION TO PRIOR DISENTANGLEMENT METHODS

While prior works such as EVP (Ji et al., 2021), EAMM (Ji et al., 2022), and PD-FGC (Wang et al., 2023a) also explore the disentanglement of geometric motion and emotion, RETA introduces a fundamentally different set of mechanisms that offer distinct advantages in practicality and performance. We compare our strategy against their core disentanglement techniques below.

**Comparison with EAMM.** EAMM (Ji et al., 2022) relies on an external emotion source video, learning to represent emotion as transferable linear displacements on unsupervised keypoints. This makes it dependent on an additional, user-provided input. In contrast, RETA is purely audio-driven, learning a dynamic emotion embedding $\boldsymbol{f}_E$ directly from the audio's prosody. Our label-free, cross-modal distillation creates a self-contained system that removes this dependency.

**Comparison with EVP.** EVP (Ji et al., 2021) operates within a unimodal framework, using an audio-to-audio cross-reconstruction objective on DTW-aligned pseudo-pairs to disentangle emotion and content entirely within the audio domain. RETA's approach is fundamentally cross-modal. By distilling knowledge from a pre-trained visual expert, we provide a richer, visually-grounded supervisory signal that is more direct than EVP's audio-only self-reconstruction.

**Comparison with PD-FGC.** PD-FGC (Wang et al., 2023a) uses statistical decorrelation to enforce feature-level independence, implicitly defining emotion as the residual information left after separating other motion factors. RETA, by contrast, employs a more direct learning signal through cross-modal knowledge distillation. We explicitly map audio prosody to a semantically rich visual emotion space, providing a stronger learning objective than relying on statistical independence.

In summary, RETA's disentanglement strategy is a significant advancement due to its unique combination of: (1) purely audio-driven emotion synthesis, eliminating the need for external references (vs. EAMM); (2) a cross-modal, visually-grounded learning paradigm that provides a richer supervisory signal than unimodal or statistical methods (vs. EVP, PD-FGC); and (3) the integration of this

advanced emotion representation with a novel, end-to-end differentiable 3DMM bridge for superior lip-sync. This holistic design enables RETA to uniquely resolve the generative trilemma.

# C    NETWORK ARCHITECTURE

## C.1    AN INTRODUCTION TO 3DMM

A 3D Morphable Model (3DMM) is a powerful statistical model of 3D facial geometry and texture, first introduced by BLANA (1999). It is constructed by applying Principal Component Analysis (PCA) to a large dataset of registered 3D face scans. The core principle of a 3DMM is that any specific face can be parametrically represented as a linear combination of a mean face and a set of basis vectors (the principal components), which capture the primary modes of variation across the training population.

The 3D shape $s_t$ of a face at time $t$, represented as a mesh of vertices, can be mathematically formulated using the notation from Section 3.1 as:

$$s_t = \bar{s} + U^{\mathrm{id}} p^{\mathrm{id}} + U^{\mathrm{exp}} p_t^{\mathrm{exp}} \tag{7}$$

where:

- $\bar{s}$ is the average face shape computed from the training dataset.
- $U^{\mathrm{id}}$ and $U^{\mathrm{exp}}$ are the orthonormal bases for identity and expression, respectively. The identity basis, $U^{\mathrm{id}}$, captures variations in facial structure that are unique to an individual (e.g., bone structure, nose shape), while the expression basis, $U^{\mathrm{exp}}$, captures non-rigid deformations corresponding to facial expressions (e.g., smiling, frowning).
- $p^{\mathrm{id}} \in \mathbb{R}^m$ is the static, low-dimensional coefficient vector that controls the identity of the face.
- $p_t^{\mathrm{exp}} \in \mathbb{R}^n$ is the dynamic, low-dimensional coefficient vector that controls the facial expression at time $t$. By manipulating these vectors, one can generate novel and realistic facial geometries.

For the purpose of talking head animation, this model is augmented to include rigid transformations, namely head pose. The pose is typically described by a rotation matrix $r \in SO(3)$ (i.e., $r$ is a 3x3 matrix that describes a valid physical rotation of the head in 3D space, without any stretching or mirror-flipping) and a translation vector $t \in \mathbb{R}^3$. Thus, the complete set of parameters to define a specific talking head instance at a moment in time is a combination of identity, expression, and pose coefficients: $(p^{\mathrm{id}}, p_t^{\mathrm{exp}}, r, t)$. More advanced 3DMMs, such as FaceVerse (Wang et al., 2022a) used in GMTalker (Xia et al., 2025), may include additional parameters for finer control over eye blinks and gaze.

The primary advantage of using a 3DMM in audio-driven talking head generation is its ability to **disentangle** the complex factors of facial appearance and motion. As demonstrated in works like SadTalker (Zhang et al., 2023b) and LES-Talker (Feng et al., 2025), this disentangled, parametric representation transforms the ill-posed, one-to-many problem of mapping audio directly to pixels into a more constrained and manageable task of predicting low-dimensional coefficients. Models can learn the strong correlation between audio and lip-related expression coefficients ($p_t^{\mathrm{exp}}$) separately from the weaker, more varied correlation with head pose ($r, t$), thereby mitigating ambiguity and enabling the generation of diverse and realistic animations. As used in RealPortrait (Ye et al., 2024), 3DMM parameters can also be used to derive appearance-agnostic motion representations (e.g., PNCC) to animate other 3D structures. In our work, we leverage this property by using the 3DMM as a differentiable regularizer to guide the learning of speech-related geometric motion.

## C.2    IMAGE ENCODER

The Image Encoder, $\mathcal{E}_I$, is responsible for extracting the complete identity information from the source image $I^i$. Its architecture is based on a standard convolutional neural network (CNN) featuring a series of downsampling residual blocks (ResBlocks) as shown in Fig 4(a). We initialize it with the pre-trained weights of the encoder from EDTalk (Tan et al., 2024a).

The encoder produces two distinct outputs designed for complementary roles in the generation process:

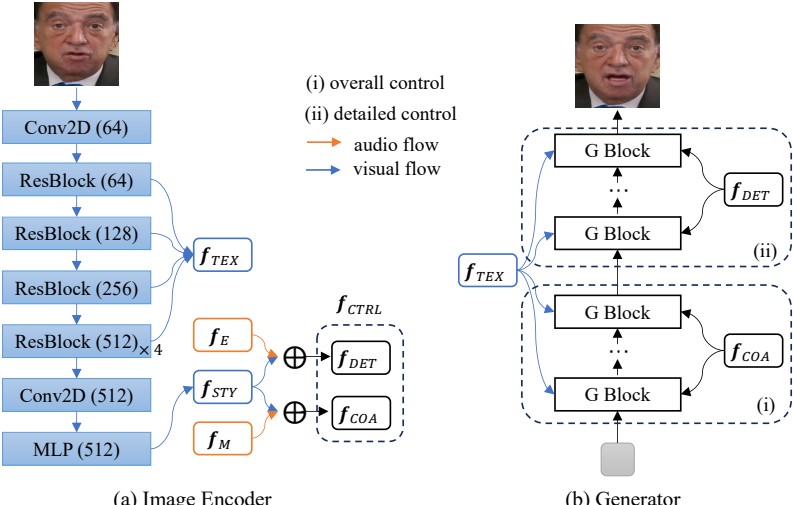

Figure 4: Network Architecture. (a) an Image Encoder, which encodes an identity image into a global style feature $f_{STY}$ and multi-scale spatial texture features $f_{TEX}$; and (b) a Generator featuring a dual-pathway structure. The first path, the **Content Pathway**, feeds the multi-scale texture features $f_{TEX}$ via skip connections. The second, the **Control Pathway**, injects the global style feature $f_{STY}$, a motion prior $f_M$, and an emotion embedding $f_E$ into the network's upsampling blocks.

- **Spatial Textural Features ($f_{TEX}$):** This is a pyramid of feature maps extracted from the intermediate layers of the encoder. Each map in the pyramid corresponds to a different spatial resolution (e.g., 64x64, 32x32). By capturing features at multiple scales, $f_{TEX}$ provides a rich, spatially-aligned representation of the source identity, including fine-grained textures and local structural details. These features are passed directly to the generator via skip connections to serve as the detailed "content" to be animated.

- **Global Style Code ($f_{STY}$):** This is a single, non-spatial vector obtained from the final, deepest layer of the encoder after pooling and flattening. It represents a high-level, global abstraction of the source image's appearance identity. Its primary role is to serve as a consistent style anchor that is injected into all modulated layers of the generator, ensuring the person's core identity remains stable across the entire animated sequence.

Together, these two feature sets provide a comprehensive representation of identity, separating the detailed, local textures from the holistic, global style.

### C.3 TEMPORAL VARIANCE MASKING.

We introduce Temporal Variance Masking (TVM), a lightweight, plug-and-play module designed to automatically identify dynamic regions within a sequence of feature maps. The core principle of TVM is that spatial regions exhibiting high variance along the temporal dimension correspond to dynamic content (e.g., a speaking mouth), while low-variance regions represent the static background.

The mask generation process, detailed in Algorithm 1, follows a multi-stage refinement pipeline. First, a variance map is generated by computing the pixel-wise variance of the feature maps $f$ across the temporal dimension, followed by an averaging operation across the channel dimension to produce a 2D map $V$. This map is then normalized to the range $[0, 1]$ and binarized using an initial threshold $\tau_{init}$ to form a coarse mask $M_{coarse}$. To enhance robustness, this mask is refined via a morphological opening operation, which effectively eliminates spurious pixel clusters not associated with coherent motion. Subsequently, a Gaussian blur is applied to smooth the contours of the refined binary mask. The final motion mask, $M_{motion}$, is obtained by resizing the smoothed map to the target dimensions and performing a final binarization with a threshold $\tau_{final}$. As illustrated in Fig. 5, the mask generated by TVM accurately isolates the primary motion areas, such as the mouth region during speech.

---

**Algorithm 1** Temporal Variance Masking

---

**Require:**

    Batch of feature maps $\boldsymbol{f} \in \mathbb{R}^{B \times C \times H \times W}$

    Initial binarization threshold $\tau_{\text{init}} \in [0, 1]$

    Morphological opening kernel size $k$

    Gaussian blur standard deviation $\sigma_g > 0$

    Final binarization threshold $\tau_{\text{final}} \in [0, 1]$

    Target output size $(H_{\text{out}}, W_{\text{out}})$

**Ensure:**

    Final motion mask $\mathbf{M}_{motion} \in \{0, 1\}^{1 \times 1 \times H_{\text{out}} \times W_{\text{out}}}$

                                     $\triangleright$ Compute channel-wise mean of temporal variance to get a 2D map

1:  $\mathbf{V} \leftarrow \text{Mean}_C(\text{Var}_B(\boldsymbol{f}))$                          $\triangleright$ Resulting $\mathbf{V}$ has shape $H \times W$

2:  $\mathbf{M}_{norm} \leftarrow (\mathbf{V} - \min(\mathbf{V}))/(\max(\mathbf{V}) - \min(\mathbf{V}))$

3:  $\mathbf{M}_{coarse} \leftarrow \mathbb{I}(\mathbf{M}_{norm} > \tau_{\text{init}})$

                                    $\triangleright$ Refine the mask using morphological opening

4:  $\mathbf{M}_{refined} \leftarrow \text{Opening}(\mathbf{M}_{coarse}, \text{kernel\_size} = k)$

                                  $\triangleright$ Smooth the mask boundaries with Gaussian blur

5:  $\mathbf{M}_{smooth} \leftarrow \text{GaussianBlur}(\mathbf{M}_{refined}, \sigma_g)$

                                    $\triangleright$ Resize and apply final binarization

6:  $\mathbf{M}_{resized} \leftarrow \text{Resize}(\mathbf{M}_{\text{smooth}}, (H_{\text{out}}, W_{\text{out}}), \text{mode='bilinear'})$

7:  $\mathbf{M}_{motion} \leftarrow \mathbb{I}(\mathbf{M}_{resized} > \tau_{\text{final}})$

8:  **return** $\mathbf{M}_{motion}$

---

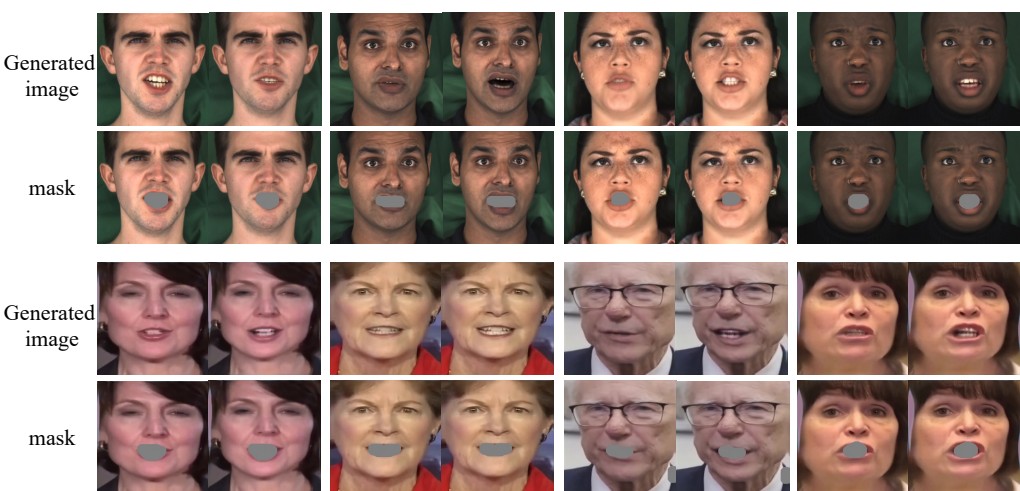

Figure 5: Visualization of the dynamic mask generated by Temporal Variance Masking module.

## C.4 GENERATOR

The Generator, $\mathcal{G}$, adapts the architecture of LIA (Wang et al., 2022b). As illustrated in Fig. 4(b), its core function is not to synthesize an image from scratch, but rather to predict a dense optical flow field and an occlusion mask. These are then used to warp the spatial features of the source image to produce the animated output. The generator skillfully integrates two distinct input streams: a content stream and a control stream.

The **content pathway** receives the pyramid of multi-scale spatial texture features, $f_{TEX}$, from the encoder via skip connections. These connections link encoder layers to their corresponding decoder layers at the same spatial resolution, providing the generator with a high-fidelity, detailed "canvas" of the source identity at every stage of the upsampling process. The generator's task is thus to realistically warp and refine this provided content rather than invent it.

The **control pathway** delivers the audio-driven animation commands. At each upsampling block in the generator, an Adaptive Instance Normalization (AdaIN) layer receives a control signal. This

signal is a combination of the global style code $f_{STY}$ and the relevant animation feature for that synthesis stage ($\boldsymbol{f}_M$ for early layers, $\boldsymbol{f}_E$ for later layers). This modulation of the generator's internal feature maps provides the precise instructions needed to generate the optical flow field and occlusion mask. The flow field dictates where each pixel of the content canvas should move, while the mask identifies newly visible regions that require inpainting during the refinement stage.

By synergizing these two pathways, the generator can efficiently warp the detailed content provided by $f_{TEX}$ according to the precise, audio-driven instructions delivered by the control signals, resulting in a high-fidelity and expressive animation that robustly preserves identity.

# D    EXPERIMENTAL SETTINGS

## D.1    DATASETS

**MEAD.** This dataset is a large-scale, high-quality laboratory dataset meticulously designed for research in audio-visual emotion recognition. It contains recordings of 60 actors (43 of whom are publicly accessible) portraying eight different emotions: anger, contempt, disgust, fear, happiness, sadness, surprise, and neutral.

**HDTF.** This dataset is a large-scale "in-the-wild" dataset curated for the task of talking head generation and audio-driven facial synthesis. Sourced from high-definition YouTube videos, it is renowned for its quality and features over 300 unique identities across a wide range of ages, genders, and ethnicities.

## D.2    IMPLEMENTATION DETAILS

For video preprocessing, we follow the methodology of EMOCA (Danecek et al., 2022) by cropping the faces and resizing the resulting videos to a $256 \times 256$ resolution for both training and testing. The dimensionality of the style code ($\boldsymbol{f}_{STY}$) and texture code ($\boldsymbol{f}_{TEX}$) is set to 512. For the loss functions, the weights for $\lambda_{L1}$ and $\lambda_{emo}$ are set to 2, while all remaining weights are set to 1. We initialize the image encoder and generator with pre-trained weights from EDTalk (Tan et al., 2024a), which is based on LIA (Wang et al., 2022b). Our audio encoder is a pre-trained Wav2Vec 2.0 model (Baevski et al., 2020). All audio is down-sampled to 16 kHz, and the learning rate is set to 0.001. The model was trained on 8 NVIDIA A100 GPUs, each with 80GB of memory.

## D.3    BASELINES

The eight baselines we compared to are summarized below:

- **EAMM.** The core of this method is decoupling and superposition. It first breaks the task into two steps: 1) generating a "base animation" that is lip-synced to the audio but has a neutral expression; and 2) extracting pure "emotional dynamics" from a separate source video and adding them as a "displacement" on top of the base animation.

- **EAT.** The core of this method is an efficient, plugin-based adaptation strategy. It freezes a pre-trained, emotion-agnostic talking-head model and inserts lightweight "emotional plugins" at various stages. These plugins alone are trained to add expressive, controllable emotions, thus avoiding costly, full-model retraining.

- **SadTalker.** The core of this method involves decoupling motion. It separately predicts 3D expression coefficients from audio and generates diverse head poses. It then bypasses 3D rendering by translating these motion parameters into 2D keypoints, which drive a 2D animation engine to synthesize the final video.

- **AniTalker.** The core of this method is to learn a universal, identity-decoupled motion space using self-supervised disentanglement. A diffusion model then generates these motion codes from audio to drive a renderer, synthesizing the final video.

- **RealPotrait.** This method uses a 'reconstruct-then-animate' pipeline. It first reconstructs a high-quality 3D head (tri-plane) from a single image. A lightweight motion adapter then computes a 'motion diff-plane' from the driving signal, which is added to the head. Finally, a specialized synthesizer generates the video.

- **KDTalker.** The core of KDTalker is that it's the first to use a Diffusion Model for predicting keypoint motion. It does not generate images directly; instead, it utilizes a custom spatiotemporal diffusion model to generate a series of dynamic changes for implicit 3D keypoints (including deformation, head pose, etc.) based on the audio. These keypoints are then fed into a separate renderer to create the final video.

- **EchoMimic.** The core of EchoMimic is a strategy of hybrid driving and collaborative training. It builds a unified diffusion model framework that simultaneously learns to be driven by both audio and facial landmarks. Through a novel training technique (e.g., randomly dropping parts of the driving signal), the model learns to generate high-quality videos whether driven by audio alone, landmarks alone, or a combination of audio and selected landmarks.

- **AniPortrait.** The core of AniPortrait is a two-stage process: first, it converts the audio signal into a sequence of 2D facial landmarks as an intermediate motion representation; then, this landmark sequence is used as a "pose controller" (similar to ControlNet) to guide a powerful pre-trained image diffusion model (like Stable Diffusion) to generate the final, high-quality animated video frame by frame.

### D.4 EVALUATION METRICS

This section provides detailed definitions for the evaluation metrics used in Section 4 to assess model performance across image quality, lip-synchronization, and emotional expressiveness. The metrics are grouped by the aspect of performance they measure.

#### D.4.1 IMAGE QUALITY AND REALISM

**PSNR (Peak Signal-to-Noise Ratio).** A widely-used metric to measure the quality of a reconstructed image by quantifying the pixel-wise difference from an original image. It is defined via the Mean Squared Error (MSE). For a ground-truth image $I$ and a generated image $\hat{I}$, both of size $H \times W$, the MSE is:

$$\text{MSE} = \frac{1}{H \times W} \sum_{i=1}^{H} \sum_{j=1}^{W} [I(i,j) - \hat{I}(i,j)]^2 \tag{8}$$

The PSNR is then defined in decibels (dB) as:

$$\text{PSNR} = 10 \cdot \log_{10} \left( \frac{\text{MAX}_I^2}{\text{MSE}} \right) \tag{9}$$

where $\text{MAX}_I$ is the maximum possible pixel value of the image (e.g., 255 for an 8-bit image). A **higher** PSNR value indicates a lower MSE and thus higher reconstruction fidelity at the pixel level.

**SSIM (Structural Similarity Index Measure).** A perceptual metric that evaluates image quality by measuring structural similarity, which correlates better with human vision than pixel-based metrics like PSNR. For two image windows $x$ and $y$, SSIM compares them based on luminance ($\mu$), contrast ($\sigma$), and structure. The formula is:

$$\text{SSIM}(x,y) = \frac{(2\mu_x\mu_y + C_1)(2\sigma_{xy} + C_2)}{(\mu_x^2 + \mu_y^2 + C_1)(\sigma_x^2 + \sigma_y^2 + C_2)} \tag{10}$$

where $\mu_x$ and $\sigma_x^2$ are the mean and variance of window $x$, $\sigma_{xy}$ is the covariance of $x$ and $y$, and $C_1, C_2$ are stabilization constants. The final score is the mean SSIM over all windows in the image. The score ranges from -1 to 1, where 1 indicates perfect similarity. A **higher** SSIM value is better.

**FID (Fréchet Inception Distance).** A metric for evaluating the quality and diversity of images from generative models Heusel et al. (2018); Seitzer (2020). It measures the Fréchet distance between the distributions of high-level image features of real images versus generated images. These features are extracted from a pre-trained Inception-v3 network and are modeled as multivariate Gaussian distributions. Given the mean and covariance of the feature distributions for real images ($m_r, C_r$) and generated images ($m_g, C_g$), the FID is:

$$d^2 = \|m_r - m_g\|_2^2 + \text{Tr}\left(C_r + C_g - 2(C_r C_g)^{1/2}\right) \tag{11}$$

where $\text{Tr}(\cdot)$ is the trace of a matrix. A **lower** FID score indicates that the distribution of generated images is more similar to the distribution of real images, suggesting better overall quality and variety.

Table 6: Quantitative comparisons with lip-sync focuse methods. ↑ denotes higher the better, ↓ denotes lower the better.

| Method | Inference | MEAD | | | | | HDTF | | | |
|--------|-----------|------|------|------|------|------|------|------|------|------|
| | | PSNR ↑ | SSIM ↑ | FID ↓ | $Sync_c$ ↑ | $ACC_e$ ↑ | PSNR ↑ | SSIM ↑ | FID ↓ | $Sync_c$ ↑ |
| Wav2Lip | 29.34 | 20.72 | 0.65 | 41.27 | **6.92** | 14.26 | 21.81 | 0.63 | 21.85 | 7.31 |
| PC-AVS | 46.24 | 16.42 | 0.47 | 46.56 | 6.21 | 13.23 | 22.14 | 0.69 | 27.33 | 6.86 |
| RETA | **55.32** | 22.19 | 0.77 | 17.09 | 5.84 | 62.35 | 22.82 | 0.75 | **14.80** | **6.21** |
| GT | - | - | - | - | 6.35 | 77.37 | - | - | - | **6.49** |

### D.4.2 LIP-SYNCHRONIZATION

**$Sync_c$ (Lip-Sync Confidence Score).** A metric that quantifies the quality of lip-synchronization by measuring the correspondence between mouth movements in a video and an audio track, as defined by the SyncNet model Chung & Zisserman (2016b). First, audio and visual (mouth) streams are embedded into a common feature space. The Euclidean distance $d(\tau) = \|v - a_\tau\|_2$ is computed between the visual embedding $v$ and the audio embedding $a_\tau$ over a window of time offsets $\tau$. The confidence score is then calculated as the difference between the median and minimum of these distances:

$$Sync_c = \text{median}(\{d(\tau)\}) - \min(\{d(\tau)\}) \tag{12}$$

A high score indicates a sharp, distinct minimum in the distance plot, signifying a strong and unambiguous temporal alignment between audio and video. A **higher** $Sync_c$ score indicates better lip-sync quality.

### D.4.3 EXPRESSIVE NATURALNESS

**$Acc_e$ (Emotion Accuracy).** The standard classification accuracy metric applied to the task of emotion recognition, as evaluated by an expert model like Emotion-FAN Meng et al. (2019). It is formally defined as the proportion of video samples in an evaluation set for which the model's predicted emotion label correctly matches the ground-truth label. For an evaluation set with $N$ samples, let $y_i$ be the true label and $\hat{y}_i$ be the predicted label for the $i$-th sample. The accuracy is:

$$Acc\_e = \frac{1}{N} \sum_{i=1}^{N} \mathbb{I}(y_i = \hat{y}_i) \tag{13}$$

where $\mathbb{I}(\cdot)$ is the indicator function. Expressed as a percentage, a **higher** $Acc\_e$ indicates that the generated expressions are more accurately perceived as the intended emotion, serving as a quantitative proxy for expressive naturalness.

## E ADDITIONAL EXPERIMENTAL RESULTS

**Qualitative Comparisons.** Fig. 6 presents a qualitative comparison between our proposed RETA and several state-of-the-art methods (SadTalker, RealPortrait, EchoMimic, and KDTalker) across the MEAD and HDTF datasets. The visual results clearly demonstrate the superiority of our method. Specifically, RETA excels in generating highly expressive emotions, achieving precise lip synchronization, and robustly preserving identity consistency, significantly outperforming all compared baselines when benchmarked against the ground truth (GT). Furthermore, to provide a comprehensive qualitative assessment, we present extensive side-by-side video comparisons using the HDTF and VFHQ datasets on our project page (https://github.com/AnonymousRETA/VisualizationRETA). These videos feature complete sentences and paragraphs, enabling a thorough evaluation of temporal consistency and expressive dynamics. These videos demonstrate how RETA's technical innovations translate to visual superiority. Our label-free emotion learning yields far more expressive results than lip-sync-focused methods like Wav2Lip, where animation is largely confined to the mouth region. Our end-to-end "differentiable bridge" avoids the error accumulation seen in GAN-based models (SadTalker, RealPortrait), producing sharper and more stable animations. Finally, our single-pass GAN generator achieves a quality competitive with diffusion models like KDTalker but at real-time speeds (>55 FPS), overcoming their critical latency issues. Thus, the qualitative results confirm RETA's ability to resolve the key trade-offs limiting prior work.

**Comparison with Emotional Generation Methods.** To further validate our claims, we directly compare RETA against emotion-driven methods like EAT and EMMN in Fig. 8. These methods face

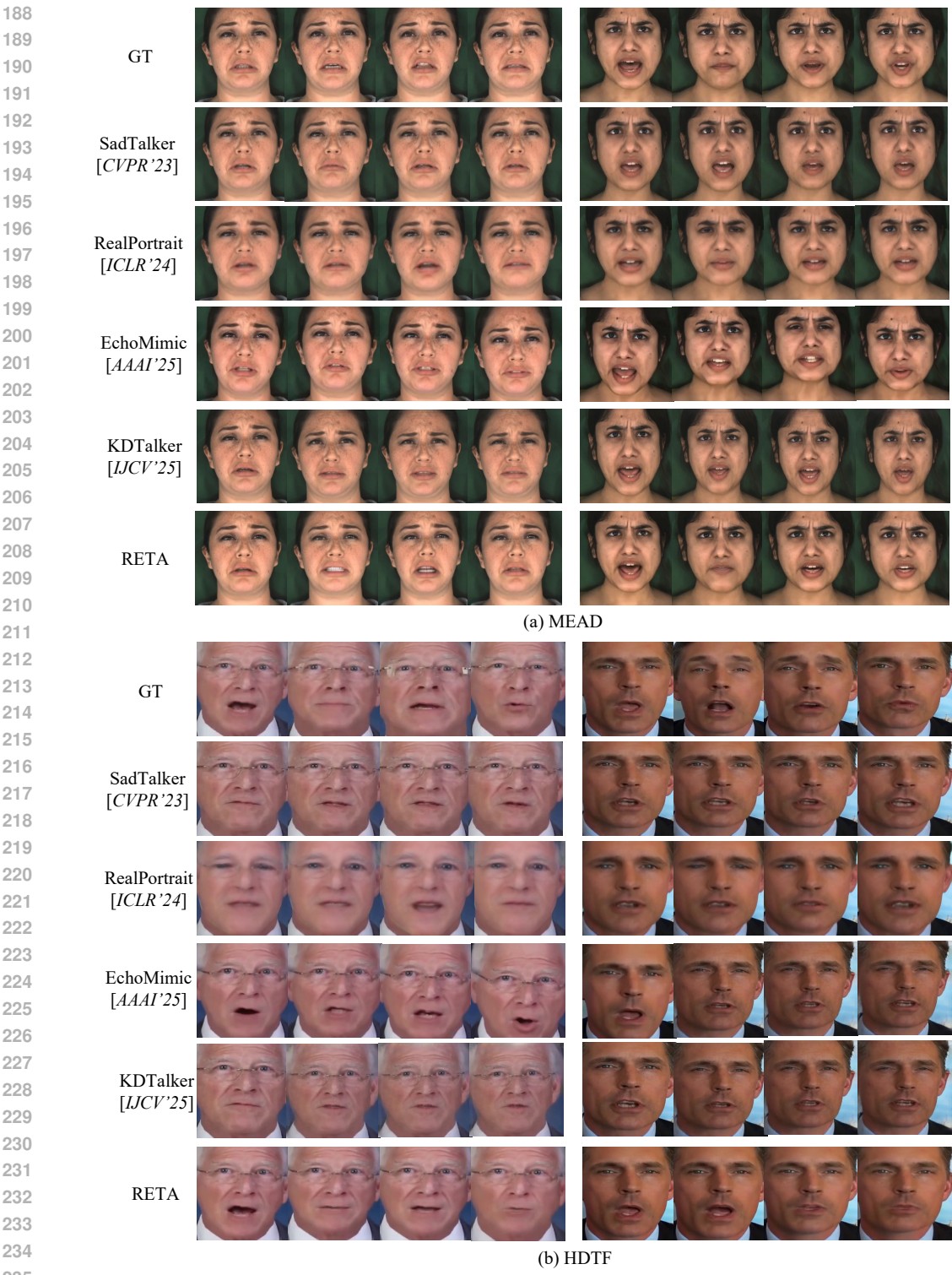

(a) MEAD

(b) HDTF

Figure 6: Qualitative comparisons.

two critical limitations that RETA overcomes. First, regarding expressiveness, their reliance on discrete labels results in static, "canned" emotions. In stark contrast, our completely label-free approach learns expressions directly from audio prosody. Second, in terms of real-time performance, they sacrifice speed for control. EAT (12.23 FPS) and EAMM (16.87 FPS) are far too slow for interactive

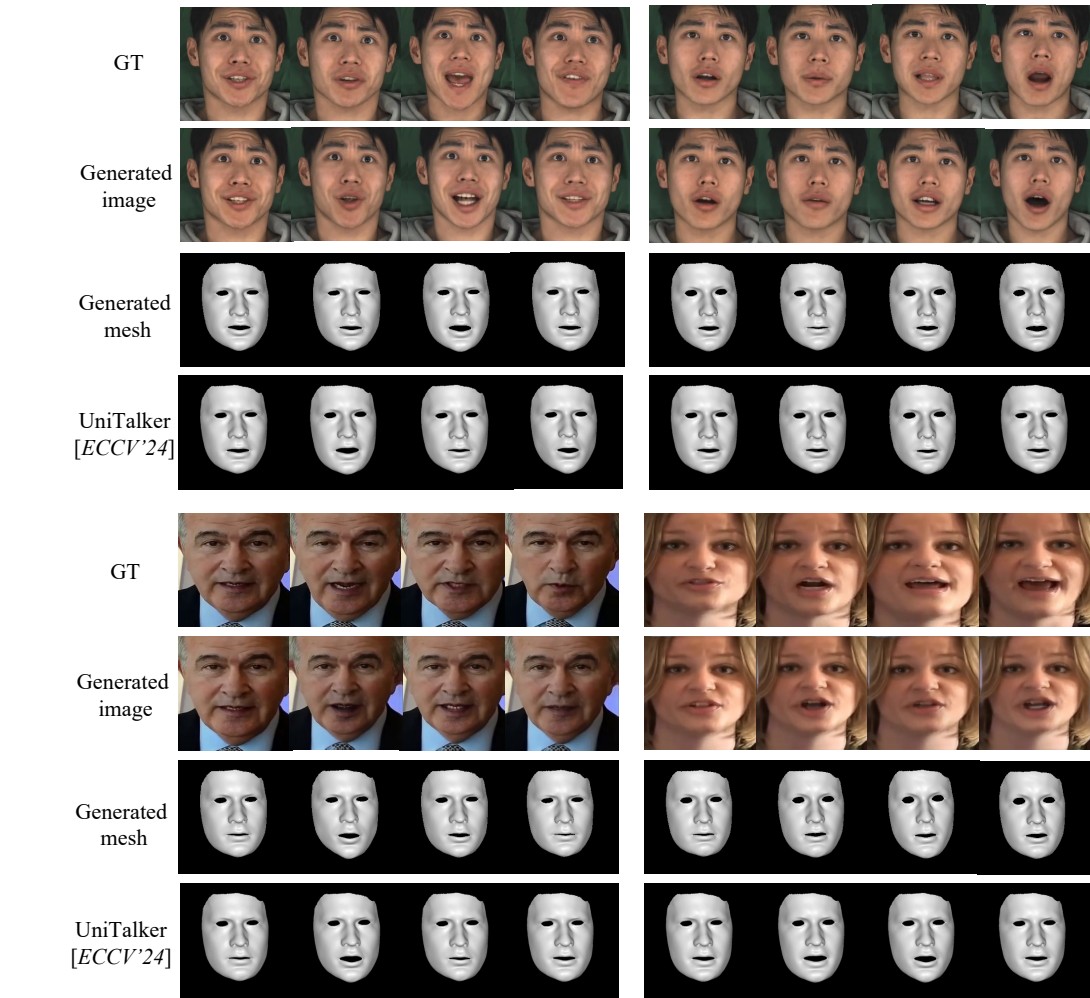

Figure 7: Visualization of Talking Head Generation and 3D Mesh Alignment.

use. RETA, however, delivers superior emotional fidelity at over 55 FPS (a 3.6x speedup). Thus, RETA is unique in resolving the core trade-off between expressiveness and real-time performance that has constrained prior work.

**Comparison with Lip-sync Focused Methods.** A direct comparison with methods hyper-focused on lip-synchronization, such as Wav2Lip (Prajwal et al., 2020a) and PC-AVS (Zhou et al., 2021), reveals the critical trade-off that RETA successfully resolves. These specialist models achieve top-tier Syncc scores by narrowly optimizing for audio-to-lip mapping, but this singular focus comes at a steep cost. By ignoring prosodic cues, they produce emotionally inert animations where motion is confined to the mouth region, resulting in extremely poor expressiveness (ACCe < 15%) and realism (FID > 21), as shown in Table 6. RETA, in contrast, avoids this severe compromise. It achieves a Syncc score (6.21 on HDTF) that is highly competitive and closely approaches the ground-truth level (6.49), while simultaneously delivering SOTA expressiveness (62.35% ACCe) and photorealism (14.80 FID). This superior balance is a direct result of our architectural design: the differentiable 3DMM bridge provides robust geometric guidance for precise lip articulation, while the parallel emotion pathway animates the entire face with nuanced expression. Thus, RETA proves that world-class expressiveness does not require sacrificing lip-sync quality, offering a holistic and far more practical solution.

**3D Mesh Alignment and Quality.** As shown in Fig. 7, the 3D mesh generated by our method maintains high fidelity with the final rendered mouth shape, validating its effectiveness as a geometric prior for the synthesis process. **Notably**, this quality is achieved without requiring any explicit 3D supervision. Forcing a model to match pre-computed, and potentially noisy, ground-truth coefficients locks in suboptimal geometric predictions early in the pipeline. These errors then propagate and are

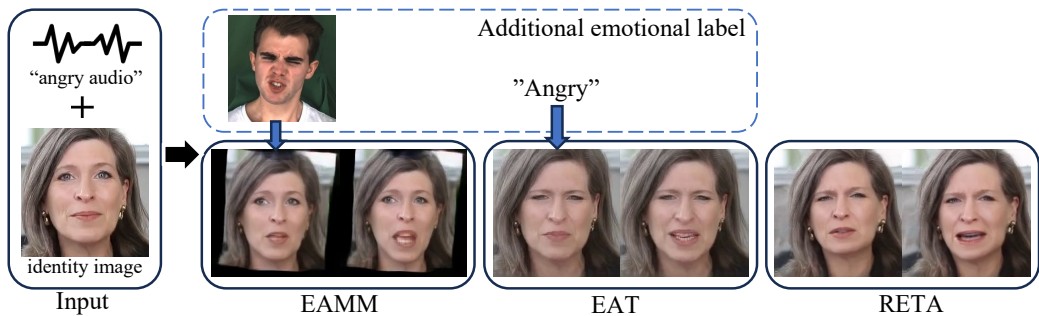

Figure 8: Comparison of Emotional Expressiveness Across Generation Methods.

amplified during the final rendering stage. Our "differentiable bridge" approach, in contrast, allows the model to learn a geometric prior that is optimally suited for the final image synthesis task, rather than merely mimicking an intermediate representation. The end-to-end backpropagation from the final image loss guides the model to discover the most effective geometric cues for generating a realistic output. Furthermore, to empirically demonstrate that our end-to-end approach still learns a geometrically sound representation, we evaluated the Lip Vertex Error (LVE) of our generated meshes against a SOTA 3D-based model (UniTalker Fan et al., 2024) as a proxy for ground-truth geometry. Our model achieves a strong LVE of 4.30, confirming that even without explicit supervision, our differentiable bridge effectively learns a high-fidelity geometric prior.

## F    COMPUTATIONAL COST ANALYSIS

To evaluate the practical applicability of RETA in real-world scenarios, this section provides a detailed analysis of its computational efficiency. The model was trained for approximately 20 hours on 8 NVIDIA A100 GPUs. All subsequent inference benchmarks were conducted on a single NVIDIA A100 GPU to ensure a fair comparison. Key performance metrics are summarized in Table 7.

- **Compared to non-diffusion-based methods:** Compared to methods employing decoupled or cascaded pipelines, such as SadTalker and EAMM, RETA's unified architecture offers significant speed advantages. These prior methods break the synthesis process into multiple discrete stages. For instance, SadTalker is bottlenecked by slow inter-process communication via disk files, while EAMM employs a complex cascade of separate modules for motion superposition and CPU-based filtering. RETA, by contrast, integrates all steps into a single, end-to-end GPU-native framework, eliminating these systemic bottlenecks. When compared to other models like EAT, their performance limitations arise from different factors. EAT relies on a computationally heavy Transformer architecture to predict its intermediate 3D keypoint representation. While this avoids a multi-stage pipeline, the complexity of the Transformer itself becomes the primary computational bottleneck. In contrast, RETA leverages a lightweight, differentiable 3DMM prior, which allows us to use a much more efficient convolutional generator. This design choice enables robust geometric guidance without the computational overhead of large-scale attention mechanisms, and the generator is further optimized for efficient batch processing.

- **Compared to diffusion-based methods:** While diffusion-based methods, such as KDTalker, have set a high bar for photorealism and expressive naturalness, they are fundamentally ill-suited for real-time applications. This limitation stems from their iterative sampling process, which requires dozens to hundreds of sequential denoising steps to generate a single frame. This sequential, multi-step inference incurs prohibitive computational costs and high latency, making them orders of magnitude slower than what is required for interactive scenarios. In stark contrast, RETA is architecturally designed for real-time performance. As a single-pass GAN-based framework, our model generates each frame in a single, efficient forward pass. This allows RETA to achieve inference speeds exceeding 55 FPS, resolving the critical trade-off between generative quality and real-time viability.

## G    EVALUATION ON GENERALIZATION AND ROBUSTNESS

Table 7: Computational profile of RETA. All inference metrics are benchmarked on a single NVIDIA A100 GPU.

| Model | VRAM (GB) | Inference Speed (FPS) | Inference Latency (ms) |
|---|---|---|---|
| EAT | 5.54 | 12.23 | 81.77 |
| SadTalker | 4.39 | 21.74 | 46.00 |
| Aniportrait | 15.84 | 3.84 | 260.42 |
| RealPortrait | 7.18 | 3.77 | 265.25 |
| KDTalker | 18.85 | 15.34 | 65.19 |
| RETA | 6.48 | 55.32 | 18.08 |

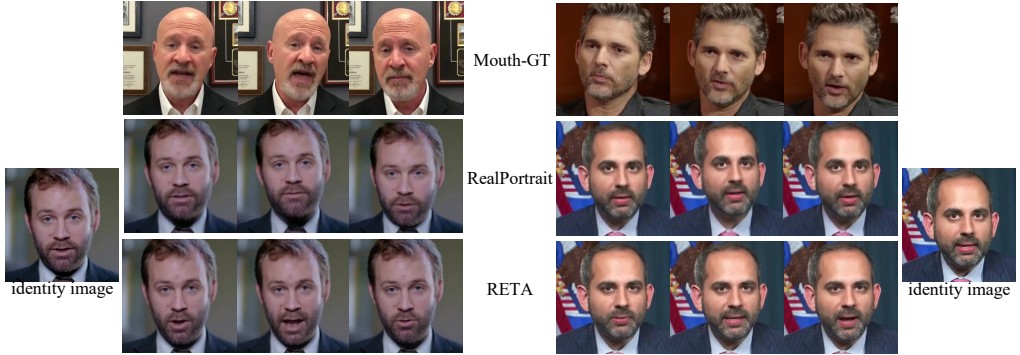

Figure 9: Qualitative Comparisons on VFHQ dataset.

To rigorously assess the real-world applicability of our framework, we evaluated RETA's zero-shot generalization and robustness on the VFHQ dataset. This "in-the-wild" video benchmark, entirely excluded from our training set, presents a challenging out-of-distribution scenario with its diverse unseen subjects and complex, noisy backgrounds.

Table 8: Generalization performance comparison on VFHQ.

| Method | PSNR | SSIM | $Sync_c$ |
|---|---|---|---|
| RealPortrait | 18.86 | 0.51 | 4.27 |
| RETA (ours) | 20.43 | 0.65 | 5.63 |

As demonstrated in Table 8, RETA achieves strong performance, quantitatively outperforming the recent GAN-based method, RealPortrait. Qualitatively, Figure 9 further illustrates that RETA maintains superior lip consistency and identity preservation even against cluttered backgrounds. This performance confirms that RETA not only animates novel identities with high fidelity but also operates robustly in unconstrained environments. This achievement on a completely unseen domain provides compelling evidence that RETA is a generalizable and robust framework well-suited for real-world deployment.

## H ABLATION STUDY

**Ablation Study Setting.** To validate the contribution of each key component, we conduct a comprehensive ablation study. The complete quantitative results are presented in Table 4, with a detailed description and analysis of each setting provided below.

- **w/o GPL:** Removes the Geometric Prior Learning (GPL) branch from the Disentanglement Feature Learning module.
- **w/o ERL:** Removes the Emotion Representation Learning (ERL) branch from the Disentanglement Feature Learning module.
- **w/o 3DMM:** Replaces the 3DMM-based geometric prior with a direct prediction from a simple decoder.
- **w/o HI:** Disables hierarchical injection, instead concatenating all features ($\boldsymbol{f}_{STY}$, $\boldsymbol{f}_M$, and $\boldsymbol{f}_E$) at the input layer of generator.

Table 9: Ablation Study of Loss Function.

| Model | PSNR | SSIM | $\text{Sync}_c$ | $\text{Acc}_e$ |
|---|---|---|---|---|
| w/o $\mathcal{L}_{syn}$ | 22.10 | 0.77 | 3.12 | 58.35 |
| w/o $\mathcal{L}_{emo}^{v-v}$ | 22.13 | 0.78 | 5.43 | 51.42 |
| w/o $\mathcal{L}_{emo}^{v-a}$ | 22.01 | 0.76 | 5.63 | 55.76 |
| Full Model | 22.19 | 0.77 | 5.84 | 62.35 |

Table 10: Sensitivity Analysis for Loss Weights.

| $\lambda_{motion}$ | $\lambda_{L1}$ | $\lambda_{emo}$ | PSNR | SSIM | $\text{Acc}_e$ |
|---|---|---|---|---|---|
| 1 | 1 | 1 | 21.89 | 0.70 | 60.07 |
| 2 | 2 | 2 | 22.10 | 0.73 | 60.72 |
| 1 | 2 | 1 | 22.15 | 0.76 | 61.21 |
| 1 | 2 | 2 | 22.19 | 0.77 | 62.35 |

- **w/o $\boldsymbol{f}_{STY}$:** Excludes the style feature $\boldsymbol{f}_{STY}$ from the generator's input.

- **w/o EDS:** Disables the Emotion Disentanglement Strategy (EDS).

- **w/o TVM:** Removes the Temporal Variance Masking (TVM) from the reconstruction loss calculation.

**Ablation Study of Loss Function.** We present an ablation study in Table 9 to demonstrate the contribution of each loss term. Removing the sync loss ($\mathcal{L}_{syn}$) degrades lip-sync performance, as evidenced by the $\text{Sync}_c$ score dropping from 5.84 to 3.12. This underscores its critical role in aligning lip movements with audio. Furthermore, the emotion loss components prove essential for expressiveness. Disabling the loss ($\mathcal{L}_{emo}^{v-v}$) and $\mathcal{L}_{emo}^{v-a}$) results in the drop in emotion accuracy, confirming their importance for visual emotion consistency. Ultimately, the full model achieves the best balance and top performance across all key metrics, validating our loss design.

**Sensitivity Analysis for Loss Weights.** We investigated the sensitivity of our model to the primary loss weights, as shown in Table 10. We found that the L1 reconstruction weight ($\lambda_{L1}$) is important for visual quality; increasing it to 2 significantly boosts both PSNR and SSIM. Concurrently, the emotion loss weight ($\lambda_{emo}$) directly enhances expressive accuracy. The best trade-off was achieved with the weights set to ($\lambda_{motion} = 1$, $\lambda_{L1} = 2$, $\lambda_{emo} = 2$), as this configuration delivered the highest performance across all evaluation metrics. Therefore, we use these values for our final model.

# I LIMITATIONS AND FUTURE WORK

**Limitations.** Our RETA framework achieves significant advancements in real-time expressive synthesis but is subject to several limitations that suggest important directions for future work. First, our label-free emotion learning strategy marks a significant step towards autonomous emotional synthesis by eliminating the need for manual annotation. However, as a distillation-based method, the semantic scope of RETA is naturally constrained by the knowledge base of the pre-trained visual teacher. This constraint implies that RETA's expressive capacity may be guided by the teacher's training characteristics, such as an emphasis on discrete emotion categories. Furthermore, RETA's performance might be influenced by potential distributional biases (e.g., demographic or cultural specificities) inherent in the teacher's original training data. Second, while our 3DMM-based geometric prior is highly effective for speech-correlated movements like jaw dynamics and lip synchronization, it is less suited for modeling dynamics that are largely independent of the audio signal, like eye blinks. This is a fundamental challenge for purely audio-driven methods, as there is no strong, reliable correlation between audio cues and blinking patterns. Attempting to force the model to learn this weak and stochastic correlation could introduce unnatural artifacts, such as erratic or rhythmically misplaced blinks, which would degrade the overall realism. Therefore, we made a conscious design choice to prioritize the model's capacity on mastering the core audio-driven tasks: precise lip synchronization and congruent emotional expression.

**Future Work.** To overcome the limitations of RETA, future research should focus on two synergistic directions for achieving full photorealistic and expressive parity with real human video. First, to mitigate the dependency on a pre-trained teacher model and expand the emotional space, a promising approach is to develop data-driven, self-supervised methods for learning expressive representations directly from large-scale, unlabelled video data. This could involve contrastive or generative models that learn a continuous latent space of expressions from audio-visual correspondences, enabling the synthesis of a richer, more dynamic, and person-specific spectrum of expressions free from categorical constraints. Second, to resolve the constrained facial dynamics inherited from the 3DMM (e.g., the absence of eye blinks), future iterations will integrate specialized modeling components. This includes complementing the 3DMM's robust guidance for large-scale movements with implicit representations or dedicated 2D dynamic sub-modules that focus on non-speech-related details in regions like the eyes.

Furthermore, the disentangled nature of our learned emotion embedding, $f_E$, provides a clear path towards explicit user control. While the current model prioritizes automatic audio-congruent expression, its modular design allows for direct manipulation of the $f_E$ vector. Future work can explore scaling this embedding to control expression intensity or replacing the audio-derived embedding entirely with one generated from other modalities, such as discrete emotion labels, text prompts, or a reference style video. This would offer users fine-grained, creative control over the final performance without altering the core generation pipeline.

## J  REPRODUCIBILITY STATEMENT

To ensure the reproducibility of our work, we provide comprehensive details regarding our code, data, and experimental setup. The source code for the RETA framework will be made publicly available upon publication. Detailed descriptions of the MEAD and HDTF datasets, along with all video and audio preprocessing steps, are provided in Appendix D. The complete network architectures for our Image Encoder, Generator, and other modules are detailed in Appendix C. All implementation details, including training hyperparameters, loss function weights, pre-trained model initialization, and the computational environment, are documented in Appendix D. To further aid in result verification, we describe the baseline methods and provide rigorous definitions for all evaluation metrics in Appendices D.3 and D.4, respectively.

## K  THE USE OF LARGE LANGUAGE MODELS (LLMS)

In this paper, we primarily utilize large language models to assist with two main tasks: refining the manuscript's writing and enhancing the layout and presentation of tables.

