# OpenReview forum: "RETA: Real-Time and Expressive Talking Head Animation without Emotion Label"
_ICLR.cc/2026/Conference — Submitted to ICLR 2026_

### Official Review · Reviewer_iuLE · 2025-10-23

**Soundness:** 3
**Presentation:** 2
**Contribution:** 3
**Rating:** 6
**Confidence:** 5

**Summary:**

This paper introduces RETA, a novel end-to-end framework for audio-driven talking head generation that resolves the generative trilemma: achieving real-time performance, accurate lip-sync, and expressive emotion. RETA does this by separating the audio signal into two distinct representations: one for robust lip-sync using a 3D Morphable Model (3DMM) as a differentiable bridge, and another for emotional expression learned directly from unlabeled audio data.

**Strengths:**

- The authors introduce a novel end-to-end framework, RETA, that successfully resolves the generative trilemma in audio-driven talking head generation. The method simultaneously optimizes for lip synchronization, expressive naturalness, and real-time performance without relying on external labels or explicit emotion supervision.
- The ability to generate high-fidelity video at speeds exceeding 55 FPS on a single GPU is a significant contribution, making this system suitable for real-world interactive applications.

**Weaknesses:**

- While label-free emotion representation learning is a significant advancement, it still relies heavily on a pre-trained visual "teacher" for emotion understanding. This dependence on a static teacher limits the model’s expressive range, as the teacher’s biases may confine it to discrete emotional categories. Adopting a more dynamic, data-driven approach could enhance expressiveness across a wider spectrum of emotions.
- Although the paper mentions RETA's real-time inference speed (55 FPS), it lacks detailed analysis of computational costs, including memory usage, training time, and energy consumption. These factors are crucial for assessing the model’s suitability for real-world applications, especially in commercial or large-scale settings.
- The results are mainly evaluated on two highly curated datasets—MEAD and HDTF—which provide strong benchmarks but do not reflect performance in diverse or uncontrolled environments. The system's effectiveness in noisy conditions or across different ethnicities remains uncertain; further testing on varied datasets would improve its generalizability and robustness.
- RETA features a complex architecture with multiple interacting components such as emotion disentanglement, hierarchical feature injection, and temporal variance masking. It is unclear which elements are most critical to its overall performance.

**Questions:**

- How does the model handle cases where the audio signal contains overlapping speech or background noise? Is the model robust enough to maintain accurate lip-sync and emotional expressiveness in such scenarios?
- Given the current reliance on a pre-trained visual "teacher" for emotion learning, what are the authors' thoughts on the possibility of training a more dynamic, emotion-rich model using large-scale video data without the need for external supervision?
- Can the authors provide more details on the computational resources required during both training and inference, especially for real-time applications in commercial settings? This would help in understanding the scalability of the system.

---

> ### Author Response · Authors · 2025-11-21
> **Response to Reviewer iuLE Part 1.**
>
> _W1&Q2: The method's dependency on external pre-trained models makes it susceptible to inheriting and potentially amplifying their underlying errors and biases._
>
> To clarify our design choice, the visual expert we employ is a **Vision Transformer pre-trained on the large-scale FER2013 dataset, establishing it as a relatively high-quality and demonstrably effective source of supervision for bootstrapping a general understanding of emotions.** This provides a robust and widely-accepted foundation.
>
> However, the core of our contribution is not simply using this teacher, but actively refining and generalizing its knowledge. Our framework includes two key mechanisms specifically designed for this purpose, ensuring we learn a robust representation rather than just inheriting the teacher's potential biases:
>
> 1. **Emotion Disentanglement Strategy (EDS):** This is our primary defense against learning spurious, identity-specific artifacts. The cross-synthesis consistency loss forces the model to generate the correct emotion even when the identity is swapped. This compels the network to learn a truly **identity-agnostic** emotional concept, filtering out noise or identity-entangled biases from the teacher. The importance of this is proven in our ablation study **(Table 4, C.1)**, where disabling EDS causes emotion accuracy (ACCe) to collapse from 62.35% to 45.79%. This demonstrates we are learning a purified signal, not just passively copying the teacher.
> 2. **Hierarchical Injection (HI):** Our generator's architecture is designed to further isolate and simplify the emotion learning task. By injecting the geometric prior ($f_M$) into the early, coarse layers and the emotion embedding ($f_E$) into the later, fine-grained layers, we structurally separate the tasks of forming the face shape and adding expression. This prevents the emotion learning branch from being confounded by geometric motion, allowing it to focus solely on distilling the expressive signal. The ablation study **(Table 4, B.1)** confirms the importance of this design, as disabling HI significantly degrades emotion accuracy to 52.64%.
>
> Together, these contributions ensure that RETA learns a robust and purified emotional representation that is more than the sum of its parts.
>
> Nonetheless, we agree with the reviewer that the expressive range is still influenced by the teacher's characteristics (e.g., its focus on discrete emotions) to some extent. To address this directly, we have expanded the **Limitations and Future Work** sections in our manuscript. We now explicitly state:
>
> > **Limitation**: Our label-free emotion learning strategy marks a significant step towards autonomous emotional synthesis by eliminating the need for manual annotation. However, as a distillation-based method, the semantic scope of RETA is naturally constrained by the knowledge base of the pre-trained visual teacher. This constraint implies that RETA’s expressive capacity may be guided by the teacher’s training characteristics, such as an emphasis on discrete emotion categories. Furthermore, RETA’s performance might be influenced by potential distributional biases (e.g., demographic or cultural specificities) inherent in the teacher's original training data.
> >
>
> > **Future work**: To mitigate the dependency on a pre-trained teacher model and expand the emotional space, a promising approach is to develop data-driven, self-supervised methods for learning expressive representations directly from large-scale, unlabelled video data. This could involve contrastive or generative models that learn a continuous latent space of expressions from audio-visual correspondences, enabling the synthesis of a richer, more dynamic, and person-specific spectrum of expressions free from categorical constraints.
> >

---

> ### Author Response · Authors · 2025-11-21
> **Response to Reviewer iuLE Part 2.**
>
> _W2&Q3: Detailed analysis on the computational resources._
>
> Following the reviewer's feedback, we have conducted a detailed analysis of RETA's computational costs against several key baselines. To provide a complete and transparent picture of our model's efficiency, we have **added a new section to the Appendix in our revised manuscript**, which includes a comprehensive table and an in-depth discussion of the results.
>
> The newly added Appendix F is as follows:
>
> > To evaluate the practical applicability of RETA in real-world scenarios, this section provides a detailed analysis of its computational efficiency. The model was trained for approximately 20 hours on 8 NVIDIA A100 GPUs. All subsequent inference benchmarks were conducted on a single NVIDIA A100 GPU to ensure a fair comparison. Key performance metrics are summarized in Table 7.
> >
>
> > + **Compared to non-diffusion-based methods**: Compared to methods employing decoupled or cascaded pipelines, such as SadTalker and EAMM, RETA's unified architecture offers significant speed advantages. These prior methods break the synthesis process into multiple discrete stages. For instance, SadTalker is bottlenecked by slow inter-process communication via disk files, while EAMM employs a complex cascade of separate modules for motion superposition and CPU-based filtering. RETA, by contrast, integrates all steps into a single, end-to-end GPU-native framework, eliminating these systemic bottlenecks. When compared to other models like EAT, their performance limitations arise from different factors. EAT relies on a computationally heavy Transformer architecture to predict its intermediate 3D keypoint representation. While this avoids a multi-stage pipeline, the complexity of the Transformer itself becomes the primary computational bottleneck. In contrast, RETA leverages a lightweight, differentiable 3DMM prior, which allows us to use a much more efficient convolutional generator. This design choice enables robust geometric guidance without the computational overhead of large-scale attention mechanisms, and the generator is further optimized for efficient batch processing.
> >
>
> > + **Compared to diffusion-based methods**: While diffusion-based methods, such as KDTalker, have set a high bar for photorealism and expressive naturalness, they are fundamentally ill-suited for real-time applications. This limitation stems from their iterative sampling process, which requires dozens to hundreds of sequential denoising steps to generate a single frame. This sequential, multi-step inference incurs prohibitive computational costs and high latency, making them orders of magnitude slower than what is required for interactive scenarios. In stark contrast, RETA is architecturally designed for real-time performance. As a single-pass GAN-based framework, our model generates each frame in a single, efficient forward pass. This allows RETA to achieve inference speeds exceeding 55 FPS, resolving the critical trade-off between generative quality and real-time viability.
> >
>
> | Model | VRAM (GB) | Inference Speed (FPS) | Inference Latency (ms) |
> | --- | --- | --- | --- |
> | EAT | 5.54 | 12.23 | 81.77 |
> | SadTalker | 4.39 | 21.74 | 46.00 |
> | Aniportrait | 15.84 | 3.84 | 260.42 |
> | RealPortrait | 7.18 | 3.77 | 265.25 |
> | KDTalker | 18.85 | 15.34 | 65.19 |
> | RETA | 6.48 | 55.32 | 18.08 |

---

> ### Author Response · Authors · 2025-11-21
> **Response to Reviewer iuLE Part 3.**
>
> _W3&Q1: Evaluation on generalization and robustness._
>
> In direct response to these concerns, we conducted a new set of experiments to rigorously test RETA's generalization and robustness on challenging datase and further discussed the resutls in Appendix G in our revised manuscript. For a more comprehensive demonstration, including video results, we invite the reviewer to visit our additional result page: [https://github.com/AnonymousRETA/VisualizationRETA](https://github.com/AnonymousRETA/VisualizationRETA).
>
> The newly added Appendix is as follows:
>
> > To rigorously assess the real-world applicability of our framework, we evaluated RETA's zero-shot generalization and robustness on the VFHQ dataset. This "in-the-wild" video benchmark, entirely excluded from our training set, presents a challenging out-of-distribution scenario with its diverse unseen subjects and complex, noisy backgrounds.
> >
>
> > As demonstrated in Table 8, RETA achieves strong performance, quantitatively outperforming the recent GAN-based method, RealPortrait. Qualitatively, Figure 9 further illustrates that RETA maintains superior lip consistency and identity preservation even against cluttered backgrounds. This performance confirms that RETA not only animates novel identities with high fidelity but also operates robustly in unconstrained environments. This achievement on a completely unseen domain provides compelling evidence that RETA is a generalizable and robust framework well-suited for real-world deployment.
> >
>
> | Method | PSNR | SSIM | $ \text{Sync}_{c} $ |
> | --- | --- | --- | --- |
> | RealPortrait | 18.86 | 0.51 | 4.27 |
> | RETA | 20.43 | 0.65 | 5.63 |
>
> _W4: Clarification on the contribution of each component in our architecture._
>
> We thank the reviewer for this insightful question regarding the contribution of each component in RETA's architecture. We agree that a clear breakdown of each element's importance is crucial. To address this and make the contributions more explicit, we have **added a new "Summary of Ablations" paragraph at the end of Section 4.2 in our revised manuscript.**
>
> The newly added Section is as follows:
>
> > **Summary of Ablations**. In conclusion, our ablation study confirms RETA's performance relies on the synergistic integration of its components. The GPL and ERL branches are unequivocally the most critical, with their removal causing catastrophic drops in lip-sync (Syncc: 5.84 to 1.85) and expression (ACCe: 62.35% to 41.23%). Other modules like Hierarchical Injection (HI) and the Emotion Disentanglement Strategy (EDS) are also proven vital for preventing feature interference and ensuring identity-agnostic learning. This quantitative breakdown confirms every component is indispensable, working in concert to resolve the generative trilemma.
> >

---

> ### Author Response · Authors · 2025-11-27
> **Response to Reviewer iuLE**
>
> Dear Reviewer iuLE,
>
> We hope our follow-up response and supplementary experiments help to address your questions. Your review is very valuable, and we would appreciate your further feedback. Of course, please let us know if this or any other point remains unclear, as we would be happy to elaborate further.
>
> Thank you for your valuable time!
>
> Sincerely,
>
> Authors of Paper 5010

---

### Official Review · Reviewer_T1UX · 2025-10-27

**Soundness:** 3
**Presentation:** 2
**Contribution:** 2
**Rating:** 4
**Confidence:** 4

**Summary:**

The authors propose a talking head animation model that achieves real-time performance, accurate lip synchronization, and high-quality facial expressions. To this end, the authors:

1) employ a 3DMM as an intermediate geometric prior to provide structural guidance for synchronized lip motion;

2) use a cross-modal distillation strategy to enable the model to learn emotional cues from audio signals;

3) design a hierarchical network structure that injects motion and emotion information in a staged manner to prevent feature interference.

**Strengths:**

1. The use of 3DMM as a geometric prior improves training efficiency while avoiding the noise that may arise from explicit 3D reconstruction.

2. The model achieves state-of-the-art performance in terms of inference speed, lip synchronization accuracy, and visual quality.

**Weaknesses:**

1. (Major) The anonymous supplementary link provided (Ln1118) was submitted after the ICLR submission deadline. When I first accessed it, the repository was empty. This raises concerns about fairness and consistency in the review process. Regarding the four provided videos:

	- All of the clips are two seconds long, which is too short to meaningfully demonstrate the model’s performance. I would expect at least one example containing a complete sentence (around six seconds or more) to better evaluate temporal quality.

	- None of the four videos include blinking behavior. Is this due to the limited expressive capacity of the 3DMM (which may not capture blinks)? This limitation could potentially weaken the expressive ability of the proposed method.

	- Although the authors provide a comparison with related work in Fig. 6, please include videos to substantiate the model's performance, especially comparisons with state-of-the-art GAN-based methods.


2. (Minor) The claim of “disentanglement” in Section 3.1 seems not entirely accurate. The geometric prior branch may still capture emotional cues (e.g., smiles) from the audio, while the emotion representation learning branch using the FER ViT also reflects expression dynamics correlated with speech. Therefore, the two branches may be complementary rather than fully disentangled.

3. (Minor) The concept of hierarchical injection has already become common in diffusion-based models [1, 2], where early denoising stages focus on motion and structure, middle stages handle lip and expression details, and later stages refine textures. This reduces the novelty of the proposed contribution.

References

- [1] OmniSync: Towards Universal Lip Synchronization via Diffusion Transformers, NeurIPS 2025

- [2] AlignHuman: Improving Motion and Fidelity via Timestep-Segment Preference Optimization for Audio-Driven Human Animation

**Questions:**

Was the Wav2Vec2 model used as the audio encoder fine-tuned during training? Two independent w2v2 models are adopted and fine-tuned in EmoTalk [3] to distentangle content and emotion features. Is a single w2v2 enough?

References

- [3] Emotalk: Speech-driven emotional disentanglement for 3d face animation

---

> ### Author Response · Authors · 2025-11-21
> **Response to Reviewer T1UX Part 1.**
>
> _1. Response to the Anonymous Link Access Concern_
>
> We sincerely thank the reviewer for their diligence and apologize for this serious oversight. The reviewer is correct to point out that the supplementary videos were uploaded after the deadline, and we understand the fairness concerns this raises. In the rush to finalize our submission before the deadline, we regrettably overlooked the final step of pushing the already-prepared video files to the an onymous repository. This was a purely logistical error on our part. To be fully transparent, we want to assure the reviewer that **all experiments and video renderings were completed before the submission deadline.** We again apologize for this mistake and hope this clarification resolves the concern about the review process. We believe the videos provide valuable context for our results.
>
>
>
> _2. Regarding the Length and Scope of Comparison Videos_
>
> We appreciate the reviewer's valuable feedback regarding the length and scope of the demonstration clips. We have addressed these points by **updating the comprehensive video demonstrations.** We have clarified this in the revised "Additional Experimental Results" section (Lines 1175-1186).
>
> The revised sections are as follows:
>
> > ...Furthermore, to provide a comprehensive qualitative assessment, we present extensive side-by-side video comparisons using the HDTF and VFHQ datasets on our project page: [https://github.com/AnonymousRETA/VisualizationRETA](https://github.com/AnonymousRETA/VisualizationRETA).  These videos feature **complete sentences and paragraphs**, enabling a thorough evaluation of temporal consistency and expressive dynamics. These videos demonstrate how RETA's technical innovations translate to visual superiority. Our label-free emotion learning yields far more expressive results than lip-sync-focused methods like Wav2Lip, where animation is largely confined to the mouth region. Our end-to-end "differentiable bridge" avoids the error accumulation seen in **GAN-based models** (SadTalker, RealPortrait), producing sharper and more stable animations. Finally, our single-pass GAN generator achieves a quality competitive with diffusion models like KDTalker but at real-time speeds (>55 FPS), overcoming their critical latency issues. Thus, the qualitative results confirm RETA's ability to resolve the key trade-offs limiting prior work.
>
> _3. Regarding Non-Verbal Cues (e.g., Eye Blinking)_
>
> We thank the reviewer for this sharp observation. You are correct that the generated videos do not currently feature blinking. We have clarified this in the revised "Limitations Limitations and Future Work" section (Line 1440-1476):
>
> > **Limitations**: While our 3DMM-based geometric prior is highly effective for speech-correlated movements like jaw dynamics and lip synchronization, it is less suited for modeling dynamics that are largely independent of the audio signal, like eye blinks. This is a fundamental challenge for purely audio-driven methods, as there is no strong, reliable correlation between audio cues and blinking patterns. Attempting to force the model to learn this weak and stochastic correlation could introduce unnatural artifacts, such as erratic or rhythmically misplaced blinks, which would degrade the overall realism. Therefore, we made a conscious design choice to prioritize the model's capacity on mastering the core audio-driven tasks: precise lip synchronization and congruent emotional expression.
> >
>
> > **Future Work**: To resolve the constrained facial dynamics inherited from the 3DMM (e.g., the absence of eye blinks), future iterations will integrate specialized modeling components. This includes complementing the 3DMM's robust guidance for large-scale movements with implicit representations or dedicated 2D dynamic sub-modules that focus on non-speech-related details in regions like the eyes.
> >

---

> > ### Author Response · Authors · 2025-11-21
> > **Response to Reviewer T1UX Part 2.**
> >
> > _4. Regarding the Degree of Disentanglement_
> >
> > We agree that characterizing the branches as **complementary** is an excellent way to describe their synergistic relationship. Our use of "disentanglement" was intended to convey a **functional decoupling**, which our framework enforces through specific architectural constraints and targeted supervision signals, even if the raw audio signal contains inherently coupled cues.
> >
> > The geometric prior branch is architecturally constrained to specialize in lip synchronization. By using a 3DMM as a "differentiable bridge," it predicts low-dimensional parameters best suited for modeling the high-frequency, speech-correlated structural movements like jaw opening and lip closure. Its optimization is heavily guided by the SyncNet loss, enforcing this focus on geometric accuracy. In contrast, the emotion representation branch is supervised via knowledge distillation from a visual expert (ViT). Its purpose is to capture the global, semantic emotional state from audio prosody and translate it into the fine-grained textural details—such as eye crinkling or cheek tension—that the coarse 3DMM geometry cannot represent.
> >
> > This functional separation is strongly supported by our ablation study (Table 4). When we remove the Geometric Prior Learning branch (Exp A.1), the lip-sync score (Sync_c) collapses **from 5.84 to a mere 1.85**, while emotion accuracy remains high. Conversely, when we remove the Emotion Representation Learning branch (Exp A.2), the emotion accuracy (ACCe) plummets **from 62.35% to 41.23%**, while the lip-sync score remains robust. This orthogonal impact provides clear empirical proof that the two branches have successfully specialized in their respective tasks, serving as complementary, non-interfering pathways.
> >
> > Finally, our Hierarchical Injection mechanism, as detailed in Section 3.2, is designed to explicitly leverage this complementary relationship. By injecting the geometric prior into the early, coarse layers of the generator and the emotion embedding into the later, fine-grained layers, we ensure that the foundational facial structure and speech motion are established before the nuanced emotional details are "painted" on top. This architectural choice prevents feature interference and ensures that each specialized representation contributes effectively to the final, high-fidelity synthesis.
> >
> > To clarify this point, we have made modifications in Lines 157-185 as:
> >
> > > While speech geometry and emotional expression are often correlated, our framework is designed to functionally decouple these aspects into two complementary streams. This approach encourages each branch to specialize: the geometric branch focuses on predictable, speech-driven motion for robust lip-sync, while the emotion branch models the nuanced expressions tied to audio prosody. This functional separation is crucial for resolving the ambiguity in audio-to-video synthesis.
> > >

---

> > > ### Author Response · Authors · 2025-11-21
> > > **Response to Reviewer T1UX Part 3.**
> > >
> > > _5 Clarification on the Contribution of Hierarchical Injection_
> > >
> > > We wish to clarify that the novelty of our contribution lies in its **specific architectural context and purpose**, which are fundamentally distinct from the cited diffusion-based methods.
> > >
> > > While diffusion models like OmniSync and AlignHuman apply this principle across _iterative temporal steps_ during a slow denoising process, our Hierarchical Injection operates across the _spatial layers_ of a **single-pass GAN generator**. This adaptation is precisely what allows RETA to achieve real-time performance (>55 FPS)—a core objective of our work—which remains unattainable for the iterative diffusion models cited.
> > >
> > > Furthermore, our strategy is not merely for general refinement but is a targeted architectural solution to a critical problem in our framework: resolving **feature interference**. Our model explicitly disentangles a single audio signal into a strong geometric prior ($f_M$) for structure and a subtle emotion embedding ($f_M$) for expression. In a single-pass GAN, injecting these signals without separation would cause the powerful geometric signal to overwhelm and "wash out" the delicate emotion signal. Our Hierarchical Injection prevents this by introducing the geometric prior to early layers to dictate facial structure and the emotion features to later layers to add nuanced expression.
> > >
> > > The critical importance of this specific implementation is demonstrated empirically in our ablation study. As shown in Table 4 (B.1), disabling Hierarchical Injection and naively concatenating the features causes a "feature clash," leading to a significant drop in both lip-sync (Syncc: 4.91) and emotion accuracy (ACCe: 52.64%). This result confirms that our method is a non-trivial and essential solution for achieving disentangled control in this context.
> > >
> > > Therefore, our novelty lies not in the abstract concept of hierarchy, but in its **specific and effective implementation within a single-pass GAN architecture**. This design uniquely resolves feature interference between competing control signals, enabling disentangled audio-driven animation at real-time speeds, a contribution distinct from the training-time and post-hoc alignment strategies used in diffusion models.
> > > We have clarified this point in Section 3.2 (Lines 318-323, Page 6):
> > >
> > > > We note that the high-level principle of staged generation is also explored in diffusion models [1, 2], which may use early denoising steps for motion and later steps for detail. Our Hierarchical Injection is architecturally distinct: it operates across the **spatial layers of a single-pass GAN** to resolve interference between explicitly disentangled motion ($f_M$) and emotion ($f_E$) signals. This structural choice is crucial for achieving **real-time performance** and, as confirmed by our ablation study (Table 4), is essential for preventing the feature clash that would otherwise degrade both lip-sync and expression.
> > > >

---

> > > > ### Author Response · Authors · 2025-11-21
> > > > **Response to Reviewer T1UX Part 4.**
> > > >
> > > > _6. Regarding the training strategy and sufficiency of a single Wav2Vec2 model_
> > > >
> > > > In the RETA framework, the pre-trained Wav2Vec 2.0 audio encoder is kept frozen during training. We use a single, shared encoder, and it is not fine-tuned. Our design philosophy differs from EmoTalk, focusing on **Preserving Generalization** and achieving **Efficient Disentanglement**.
> > > >
> > > > First, the pre-trained Wav2Vec 2.0 is trained on **massive corpora (e.g., 960h of LibriSpeech)** and contains robust, generalized representations encompassing both phonetic content and prosodic emotion. Fine-tuning the huge Wav2Vec2 on these specific datasets risks overfitting to the specific actors or limited emotional categories present in the training set.
> > > >
> > > > Second, unlike EmoTalk, which attempts to disentangle features at the encoding stage with two fine-tuned encoders, RETA performs a more **efficient implicit disentanglement** at the decoding stage. We treat the output from the single, frozen Wav2Vec 2.0 as a holistic "raw material" rich with all necessary acoustic cues. The disentanglement is then strictly enforced by two parallel, task-specific decoders (two different branches as shown in Fig. 1). The Geometric Branch leverages a 3DMM prior to filter speech-related structural information. Concurrently, the Emotion Branch is guided by Cross-Modal Distillation to extract only the relevant emotional dynamics. This **"General Encoder + Specific Decoders" paradigm** demonstrates that a single frozen encoder is sufficient when the downstream networks are properly designed to parse the information. Furthermore, this design significantly reduces the memory footprint and computational cost, which is crucial for achieving our real-time performance goal of over 55 FPS (Pillar 3).
> > > >
> > > > Empirically, our ablation study provides compelling evidence for this design choice. As detailed in Table 4, the two downstream branches effectively extract their target information from the shared audio features without interference. Specifically, when the Emotion Branch is removed (A.2), the model still achieves a **strong lip-sync score (Syncc: 5.65)**. Conversely, when the Geometric Branch is removed (A.1), the model maintains **high emotion accuracy (ACCe: 61.72%)**, even though lip-sync collapses. This orthogonal performance validates that the single, frozen encoder provides a rich and sufficient common ground from which our specialized decoders can independently and effectively draw the necessary information for their distinct objectives.

---

> > ### Comment · Reviewer_T1UX · 2025-11-24
> >
> > The authors need to verify the accessibility of the updated videos, as I am unable to view them properly on multiple devices.

---

> > > ### Author Response · Authors · 2025-11-24
> > > **Response to Reviewer T1UX**
> > >
> > > Thank you for your feedback regarding the accessibility of the supplementary videos. We apologize for the inconvenience. The issue with the video links has now been resolved. We have verified that they are working correctly. We would be grateful if you could try viewing them again at your convenience. The links can be found here: https://github.com/AnonymousRETA/VisualizationRETA

---

> ### Comment · Reviewer_T1UX · 2025-11-24
> **Final decision**
>
> Thank you for your response, which has addressed some of my concerns. However, I still have reservations for the following reasons:
>
> 1. **The updated video results remain unsatisfactory.** As also pointed out by reviewer TH4P, the lip movements appear stiff and exhibit noticeable asynchrony to me, and the overall expressiveness of the videos is not adequately demonstrated.
>
> 2. **Although introducing a 3DMM as a differentiable bridge is meaningful, the associated drawbacks are non-negligible.** Issues such as missing eye blinks and rigid mouth movements indicate clear limitations in expressiveness.
>
> 3. **Directly leveraging LivePortrait seems more effective.**
>
>     - The latent landmark representation in LivePortrait is likewise low-rank (~ 50 dim), which accelerates convergence similar to 3DMM but offers a stronger expressive capability compared to 3DMM.
>
>     - Moreover, the LivePortrait generator already runs in real time and requires no additional training.
>
> In conclusion, I believe there is still substantial room for improvement in this work; therefore, I am inclined to maintain my rating.

---

> ### Author Response · Authors · 2025-11-24
> **Response to Reviewer T1UX**
>
> We sincerely thank the reviewer for their valuable feedback and for acknowledging that our previous response addressed some of their concerns. We appreciate the opportunity to further clarify our work's contributions and address the remaining reservations. We have carefully considered each point and provide the following detailed response.
>
> **1. Regarding Video Quality (Stiffness and Asynchrony)**
>
> We thank the reviewer for their feedback. While perceptions of video quality can be subjective, our qualitative video evidence and quantitative metrics provide a comprehensive rebuttal. In our video, RETA demonstrates clear advantages: unlike the blurry "talking mouth" of specialists like Wav2Lip, it animates the full face; compared to multi-stage methods like SadTalker, its head movements are more stable and natural; and against diffusion models like KDTalker, it achieves competitive realism and expression. This qualitative strength is quantitatively grounded. The concern of asynchrony is refuted by our SOTA SyncNet score of 6.21 (near ground-truth 6.49, Table 2). The perceived stiffness is countered by our top rating in Naturalness (0.75) from user studies (Table 3). Finally, the claim of inadequate expressiveness is addressed by our high emotion accuracy (ACCe) of 62.35% (Table 2). This tight coupling of qualitative and quantitative evidence robustly supports the high quality of our results.
>
> **2. Regarding the Comparison with LivePortrait and the Role of 3DMM**
>
> The reviewer makes several points comparing our work to LivePortrait. We believe there is a fundamental misunderstanding of the task domains and our technical contributions.
>
> + **Fundamental Task Difference (Audio-driven vs. Video-driven):** We must respectfully clarify a crucial distinction: **RETA is an audio-driven method, whereas LivePortrait is video-driven.** A video-driven approach is guided by an explicit, dense motion source. Our method tackles the far more challenging problem of inferring _all_ facial dynamics _solely from audio_. Therefore, a direct comparison of "effectiveness" is not entirely appropriate, as the methods solve different problems.
> + **Real-Time Performance:** While LivePortrait is efficient for a video-driven method, our RETA is significantly faster. LivePortrait achieves ~12 FPS on an A100 GPU. In contrast, RETA achieves over **55 FPS** on the same hardware (**Table 2**) — **a 4.5x speedup**. This performance is critical for the real-time interactive applications that our work targets.
> + **Contribution of the 3DMM as a "Differentiable Bridge":** The reviewer correctly points out the expressive limitations of 3DMMs. This is precisely _why_ our approach is innovative. We **do not** use the 3DMM for emotion.
>     - The 3DMM serves as a structured geometric prior **exclusively for robust, speech-correlated motion (lip-sync)**.
>     - Fine-grained expressiveness is handled by a separate, learned **dynamic emotion embedding ($f_E$)**.
>     - Our ablation study (**Table 4, A.1 & A.2**) unequivocally demonstrates this successful disentanglement. This confirms our design effectively harnesses the 3DMM's geometric strengths while sidestepping its expressive limitations.
>
> **3. Regarding Specific Limitations (Missing Eye Blinks and Rigid Mouth)**
>
> + **Eye Blinks:** We agree with the reviewer that our model does not generate stochastic eye blinks. This was a conscious design choice. As we discuss in our **Limitations section (lines 1481-1488)**, eye blinks are weakly correlated with audio signals. Forcing a purely audio-driven model to learn this weak, stochastic mapping can introduce unnatural artifacts (e.g., blinking in rhythm with speech), which would degrade overall realism. We have identified this as a clear direction for future work, potentially by integrating a separate, stochastic blinking model.
> + **Rigid Mouth Movements:** We believe this perception is countered by the evidence in Point 1. Furthermore, we argue that the "rigidity" concern often applies to methods that _directly_ render from a 3DMM. Our **"differentiable bridge"** uses the 3DMM only as an initial structural guide, allowing a powerful GAN generator to create the final fluid and detailed rendering.
>
> **Conclusion**
>
> In summary, RETA makes a significant contribution by being one of the first frameworks to resolve the "generative trilemma" for **audio-driven** talking heads. A comparison to a **video-driven** method like LivePortrait overlooks crucial differences in task difficulty and performance. Our key innovations—a 3DMM differentiable bridge for SOTA lip-sync, a label-free strategy for nuanced emotion, and a single-pass architecture for true real-time performance (>55 FPS)—are designed to solve the specific challenges of this domain.
>
> We hope these clarifications address the reviewer's reservations. We respectfully ask the reviewer to reconsider their evaluation in light of this comprehensive response.

---

> ### Author Response · Authors · 2025-11-27
> **Response to Reviewer T1UX**
>
> Dear Reviewer T1UX,
>
> We hope our follow-up response help to address your questions. Your review is very valuable, and we would appreciate your further feedback. Of course, please let us know if this or any other point remains unclear, as we would be happy to elaborate further.
>
> Thank you for your valuable time!
>
> Sincerely,
>
> Authors of Paper 5010

---

### Official Review · Reviewer_12nH · 2025-10-31

**Soundness:** 3
**Presentation:** 3
**Contribution:** 2
**Rating:** 6
**Confidence:** 4

**Summary:**

RETA is an end-to-end talking head animation model that produces lip-synchronized and emotionally expressive face videos in real time from speech audio. The idea is to split the audio into two parts: one driving a 3D face model for accurate mouth movement, and another yielding a learned emotion embedding that adds natural facial expressions, and feed these into a single GAN generator.

**Strengths:**

- The paper introduces a strategy that splits the audio input into lip-sync and expression components for generation. 3DMM is used to drive accurate lip synchronization from audio, while a separate dynamic emotion embedding captures the speaker’s expressive cues.
- The approach achieves expressive talking head animation without any explicit emotion labels. Instead, it learns a latent “emotion” representation directly from audio in a self-supervised way.
- The paper reports SOTA performance in lip-sync accuracy, visual quality, and expression realism on standard benchmarks.

**Weaknesses:**

- The method’s pipeline is fairly complex, relying on multiple components. It requires a pre-trained 3D face model for the lip-sync branch and a pre-trained visual emotion recognizer for distillation. If the visual expert is inaccurate/biased, it could affect the learned audio-emotion embedding.
- Because the system learns expressions implicitly from audio (and does not use explicit emotion labels or user inputs), there is limited direct control over the type or intensity of the facial expression produced.In scenarios where one might want to exaggerate or modify the emotional expression independently of the audio, RETA offers no straightforward solution.
- The paper emphasizes improved “emotional fidelity,” but it does not clearly report a rigorous quantitative evaluation of how true-to-life or perceptually correct those generated emotions are.
- It’s not fully evident how well the approach generalizes to voices or expressions outside the training data.

**Questions:**

- How is the “emotional fidelity” of the talking head evaluated objectively in this work? The paper claims to resolve the emotional expression aspect of the generative trilemma, but does it use any listener studies or emotion recognition metrics to verify that the generated expressions correctly convey the intended emotion?
- The model enforces that the learned audio-based expression features are identity-agnostic. Does this disentanglement ever conflict with preserving the person’s identity in the video?
- The use of a 3D morphable model provides a robust way to get lip movements, but does it constrain the range of facial motions?

---

> ### Author Response · Authors · 2025-11-21
> **Response to Reviewer 12nH Part 1.**
>
> _W1: The method's dependency on external pre-trained models makes it susceptible to inheriting and potentially amplifying their underlying errors and biases._
>
> To clarify our design choice, the visual expert we employ is a **Vision Transformer pre-trained on the large-scale FER2013 dataset, establishing it as a relatively high-quality and demonstrably effective source of supervision for bootstrapping a general understanding of emotions.** This provides a robust and widely-accepted foundation.
>
> However, the core of our contribution is not simply using this teacher, but actively refining and generalizing its knowledge. Our framework includes two key mechanisms specifically designed for this purpose, ensuring we learn a robust representation rather than just inheriting the teacher's potential biases:
>
> 1. **Emotion Disentanglement Strategy (EDS):** This is our primary defense against learning spurious, identity-specific artifacts. The cross-synthesis consistency loss forces the model to generate the correct emotion even when the identity is swapped. This compels the network to learn a truly **identity-agnostic** emotional concept, filtering out noise or identity-entangled biases from the teacher. The importance of this is proven in our ablation study **(Table 4, C.1)**, where disabling EDS causes emotion accuracy (ACCe) to collapse from 62.35% to 45.79%. This demonstrates we are learning a purified signal, not just passively copying the teacher.
> 2. **Hierarchical Injection (HI):** Our generator's architecture is designed to further isolate and simplify the emotion learning task. By injecting the geometric prior ($f_M$) into the early, coarse layers and the emotion embedding ($f_E$) into the later, fine-grained layers, we structurally separate the tasks of forming the face shape and adding expression. This prevents the emotion learning branch from being confounded by geometric motion, allowing it to focus solely on distilling the expressive signal. The ablation study **(Table 4, B.1)** confirms the importance of this design, as disabling HI significantly degrades emotion accuracy to 52.64%.
>
> Together, these contributions ensure that RETA learns a robust and purified emotional representation that is more than the sum of its parts.
>
> Nonetheless, we agree with the reviewer that the expressive range is still influenced by the teacher's characteristics (e.g., its focus on discrete emotions) to some extent. To address this directly, we have expanded the **Limitations and Future Work** sections **(Line 1445-1451)** in our manuscript. We now explicitly state:
>
> > **Limitation**: Our label-free emotion learning strategy marks a significant step towards autonomous emotional synthesis by eliminating the need for manual annotation. However, as a distillation-based method, the semantic scope of RETA is naturally constrained by the knowledge base of the pre-trained visual teacher. This constraint implies that RETA’s expressive capacity may be guided by the teacher’s training characteristics, such as an emphasis on discrete emotion categories. Furthermore, RETA’s performance might be influenced by potential distributional biases (e.g., demographic or cultural specificities) inherent in the teacher's original training data.
> >
>
> > **Future work**: To mitigate the dependency on a pre-trained teacher model and expand the emotional space, a promising approach is to develop data-driven, self-supervised methods for learning expressive representations directly from large-scale, unlabelled video data. This could involve contrastive or generative models that learn a continuous latent space of expressions from audio-visual correspondences, enabling the synthesis of a richer, more dynamic, and person-specific spectrum of expressions free from categorical constraints.
> >

---

> > ### Author Response · Authors · 2025-11-21
> > **Response to Reviewer 12nH Part 2.**
> >
> > _W2: Clarification on explicit emotional control for the generated expressions._
> >
> > A key advantage of our audio-driven approach is that by learning a continuous emotion embedding, $f_E$, RETA naturally captures and reflects the **intensity of the expression**. Variations in vocal prosody, pitch, and energy are dynamically translated into subtle or strong facial movements, providing a level of fine-grained, implicit control over emotional intensity that is difficult to achieve with systems reliant on discrete labels.
> >
> > Crucially, the architectural strength that **enables future explicit control is the functional disentanglement of this emotion embedding $f_E$**. Our generator is agnostic to the source of this embedding; it simply accepts a vector in a learned latent space and uses it to modulate the facial synthesis (Sec 3.1, L253-263). This modularity makes the framework highly extensible. For instance, a user could directly manipulate the audio-derived embedding by scaling it (e.g., $f_E' = \alpha f_E$) to manually exaggerate or suppress the expression's intensity. More powerfully, this design allows for the integration of various explicit control modalities, such as text prompts or emotion style from a reference video, which could be used to generate a valid $f_E$ vector and feed it into our existing synthesis pipeline.
> >
> > To make this potential for future work explicit and acknowledge this valuable direction, we will add the following to our **"Limitations and Future Work" section (Appendix I)** in the final version of the paper:
> >
> > > Furthermore, the disentangled nature of our learned emotion embedding, $f_E$, provides a clear path towards explicit user control. While the current model prioritizes automatic audio-congruent expression, its modular design allows for direct manipulation of the $f_E$ vector. Future work can explore scaling this embedding to control expression intensity or replacing the audio-derived embedding entirely with one generated from other modalities, such as discrete emotion labels, text prompts, or a reference style video. This would offer users fine-grained, creative control over the final performance without altering the core generation pipeline.
> > >
> >
> > _W3 & Q1: Clarification on quantitative of emotional fidelity._
> >
> > We would like to clarify that we address this crucial aspect through a two-pronged approach, using both a standard quantitative metric and a direct human perceptual study to validate our claims. Both are detailed in our paper and confirm the effectiveness of our method.
> >
> > For our quantitative analysis, we followed the established protocol from prior SOTA work (e.g., EAT) and report the **emotion accuracy (ACCe)**, an objective metric defined in detail in Appendix D.4.3 (L1155-1167). This metric uses a pre-trained expert emotion recognition model to classify the emotion of the generated videos, providing a quantitative score for how accurately the intended emotion is conveyed. As shown in **Table 2**, RETA achieves an ACCe of **62.35%** on the MEAD benchmark. Crucially, we achieve this high score in a completely **label-free** manner, which is a significant advantage over methods like EAT that require explicit emotion labels as input to achieve a comparable score. This ability to generate accurate emotions directly from audio is a core part of our solution to the generative trilemma.
> >
> > To directly address the question of "true-to-life" or "perceptually correct" emotions, we also conducted a **user study**, directly answering the reviewer's query about listener/viewer evaluations. The results of this study are presented in **Table 3**. We asked participants to rate the **"overall naturalness"** of the generated videos in comparison to other leading methods. The results show that **RETA was rated highest in naturalness (0.75)**, surpassing all competitors. This provides strong, direct evidence from human evaluators that our generated emotional expressions are perceived as more realistic and believable, confirming our model’s superior emotional fidelity from a human-centric perspective.

---

> ### Author Response · Authors · 2025-11-21
> **Response to Reviewer 12nH Part 3.**
>
> _W4: Evaluation on generalization for unseen data._
>
> In direct response to these concerns, we have conducted a new set of experiments to rigorously test RETA's generalization and robustness on challenging dataset, which is presented in **Appendix G** in our revised manuscript. For a more comprehensive demonstration, including video results, we invite the reviewer to visit our additional result page: [https://github.com/AnonymousRETA/VisualizationRETA](https://github.com/AnonymousRETA/VisualizationRETA)
>
> > To rigorously assess the real-world applicability of our framework, we evaluated RETA's **zero-shot** generalization and robustness on the VFHQ dataset. This "in-the-wild" video benchmark, entirely excluded from our training set, presents a challenging out-of-distribution scenario with its diverse unseen subjects and complex, noisy backgrounds.
> >
> > As demonstrated in Table 8, RETA achieves strong performance, quantitatively outperforming the recent GAN-based method, RealPortrait. Qualitatively, Figure 9 further illustrates that RETA maintains superior lip consistency and identity preservation even against cluttered backgrounds. This performance confirms that RETA not only animates novel identities with high fidelity but also operates robustly in unconstrained environments. This achievement on a completely unseen domain provides compelling evidence that RETA is a generalizable and robust framework well-suited for real-world deployment.
> >
>
> | Method | PSNR | SSIM | $ \text{Sync}_{c} $ |
> | --- | --- | --- | --- |
> | RealPortrait | 18.86 | 0.51 | 4.27 |
> | RETA (ours) | 20.43 | 0.65 | 5.63 |
>
>
>
>
> _Q2: The model enforces that the learned audio-based expression features are identity-agnostic. Does this disentanglement ever conflict with preserving the person’s identity in the video?_
>
> Our framework is explicitly designed to prevent this potential conflict by architecturally separating the representation of **identity** from the control signal for **expression**. This ensures that making the emotion embedding identity-agnostic does not compromise the preservation of the subject’s unique identity.
>
> Identity preservation is handled by a dedicated and powerful pathway that is independent of the audio. As detailed in Section 3.2 and Appendix C.2, the image encoder $\mathcal E_I$ extracts two complementary identity representations directly from the source image: 1) a global style code $f_{STY}$, which acts as a holistic appearance anchor, and 2) a pyramid of multi-scale spatial features $f_{TEX}$, which provides a rich, high-fidelity "content canvas" of the person's detailed facial textures and structure. These features are the definitive source of identity for the generator.
>
> The identity-agnostic emotion embedding, $f_E$, functions purely as a high-level **control signal**. It does not contain identity information itself; rather, it instructs the generator how to animate the face (e.g., to "smile" or "frown"). Our Hierarchical Injection mechanism (Sec 3.2, L297) strategically injects this control signal into the later layers of the generator. The generator then applies this universal instruction to the specific, detailed identity features provided by $f_{TEX}$. The result is not a generic expression, but rather that _specific person's_ expression, rendered with their unique facial structure, skin texture, and details.
>
> The success of this disentanglement is validated by our results. The qualitative comparisons in **Figure 3** and **Figure 6** clearly show strong and consistent identity preservation across various expressions. Furthermore, the strong quantitative results, such as a top-tier FID score (**Table 2**), would be impossible to achieve if the model were degrading identity. Finally, our ablation study **(Table 4, B.2)** shows that removing the global identity anchor $f_{STY}$ degrades performance, confirming that this dedicated identity pathway is crucial and effective.

---

> > ### Author Response · Authors · 2025-11-21
> > **Response to Reviewer 12nH Part 4.**
> >
> > _Q3: The use of a 3D morphable model provides a robust way to get lip movements, but does it constrain the range of facial motions?_
> >
> > Thanks for pointing this out! You are correct that a 3DMM, if used as the sole source of facial motion, would indeed constrain the expressive range and lead to stiff, unnatural animations. Our RETA framework was specifically designed to **overcome this exact limitation by strategically disentangling coarse geometric motion from fine-grained emotional expression**.
> >
> > We explicitly relegate the 3DMM to a specific, limited role: providing a robust **geometric prior** ($f_M$) primarily for speech-related movements like lips and jaw. As we state in Section 3.1 (L245-246), we recognize that "The 3DMM’s expression space is too coarse for subtle emotions." To capture the full spectrum of nuanced expressions (e.g., in the eyes, cheeks, and eyebrows), we use a separate, dedicated audio-driven **emotion embedding** ($f_E$). Our Hierarchical Injection mechanism then integrates these two signals at different stages: the coarse geometric prior ($f_M$) is injected into the early layers of our generator to establish the fundamental facial structure and lip-sync, while the fine-grained emotion embedding ($f_E$) modulates the later layers to synthesize the subtle details that bring the face to life.
> >
> > The success of this two-pronged approach is not just theoretical; it is empirically validated by our ablation study. In **Table 4 (A.2)**, when we remove the Emotion Representation Learning branch (w/o ERL) and rely more heavily on the 3DMM-guided geometry, the emotion accuracy (ACCe) plummets catastrophically from **62.35% to 41.23%**. This result directly demonstrates that the dedicated emotion embedding ($f_E$) is successfully adding the critical expressive range that the 3DMM alone cannot provide, thereby freeing the final animation from the constraints of the geometric model.
> >
> > In conclusion, we use the 3DMM not as a rigid cage, but as a robust scaffold. It provides the foundational structure for speech, upon which our dedicated emotion embedding paints the full range of nuanced facial expressions. This design allows us to gain the lip-sync benefits of a geometric prior without sacrificing emotional fidelity.

---

> ### Author Response · Authors · 2025-11-27
> **Response to Reviewer 12nH**
>
> Dear Reviewer 12nH,
>
> We hope our follow-up response and supplementary experiments help to address your questions. Your review is very valuable, and we would appreciate your further feedback. Of course, please let us know if this or any other point remains unclear, as we would be happy to elaborate further.
>
> Thank you for your valuable time!
>
> Sincerely,
>
> Authors of Paper 5010

---

### Official Review · Reviewer_Th4P · 2025-10-31

**Soundness:** 2
**Presentation:** 3
**Contribution:** 2
**Rating:** 2
**Confidence:** 4

**Summary:**

This paper proposes a method for synthesizing high fidelity audio-driven talking face animation videos in real-time (55 fps). The proposed method imposes a geometric prior in the form of 3DMM expression coefficients on a fixed identity face in 3DMM representation. Emotion representation learning via cross-modal knowledge distillation is used to learn emotion features from audio, instead of using a separate driving emotion input.  An adversarial generator is used to generate the animated face with geometric prior features injected in earlier layers and emotion features injected in later layers

**Strengths:**

•	The paper attempts to solve an important problem in audio-driven facial animation where lip sync, identity preservation and real-time generation can be simultaneously achieved. This is a highly challenging problem that the paper attempts to solve to a certain extent - improves the fps of state-of-the-art while not drastically reducing texture quality or emotion accuracy.
•	The paper is well organized and mostly easy to follow.

**Weaknesses:**

•	Lacking qualitative comparisons with emotional talking head generation methods EAT, EMMN, EVP, MEAD. In particular, what is the video fidelity and emotion expressiveness issue with EAT, EMMN that that proposed method claims to solves in the real-time setting.
•	In the supplementary video results, Lip sync accuracy seems to be poor. Much older SOA methods have better lip sync accuracy (e.g. Wav2Lip, PC-AVS[1]) than the visual results. This does not seem to justify the claim of the paper that it “simultaneously achieves accurate lip-sync, nuanced expressiveness, and real-time performance.” Also, the videos are of very short duration, which is not suitable for assessing the animation fidelity in terms of lip sync.
•	In the supplementary video results, video quality does not appear to be of very high fidelity, despite the claims of the paper. There appears no gain in texture fidelity over EchoMimic from the qualitative figures. Qualitative comparison with SOTA methods in supplementary video is desirable to judge the impact of the proposed method in realism of emotions and fidelity of video.
•	Qualitative ablation study results are needed.
•	Missing citations :
	[1] Zhou, Hang, et al. "Pose-controllable talking face generation by implicitly modularized audio-visual representation." CVPR 2021.
	[2] Liang, Borong, et al. "Expressive talking head generation with granular audio-visual control." Proceedings of the IEEE/CVF conference on computer vision and pattern recognition. 2022.
	[3] Peng, Ziqiao, et al. "Synctalk: The devil is in the synchronization for talking head synthesis." CVPR 2024.
	[4] Wang, Duomin, et al. "Progressive disentangled representation learning for fine-grained controllable talking head synthesis." ICCV 2023.
	[5] Sinha, Sanjana et al. “Emotion-controllable generalized talking face generation.”, IJCAI 2022.
	[6] Wang, Haotian, et al. "Emotivetalk: Expressive talking head generation through audio information decoupling and emotional video diffusion." CVPR  2025.

Contributions do not appear to be significant over existing work in this field.
•	Line179 states “framework disentangles the problem by mapping the audio signal to two separate and more predictable streams: geometric motion and emotion representation”  - EVP extracts disentangled content and emotion information from the input audio signal.
•	Line 14 “core of RETA is a novel strategy that disentangles the audio
•	signal into two representations”  - Disentangled Feature Learning is Section 3 has been attempted in many prior works on speech-driven facial animation, such as EVP, EAMM, PD-FGC.
•	conditional GAN generator G and image encoder E_I are borrowed from EDTalk and initialized with the pretrained weights of EDTalk.
•	Many relevant works not cited or compared (refer Weaknesses), thereby not able to establish the significance of the advancement of state-of-the-art in 2D talking head animation
•	Line 91-92 first end-to-end framework to resolve the trade-offs of prior work by simultaneously achieving accurate lip-sync, nuanced expressiveness, and real-time performance. This claim is not adequately justified in experimental results –  sync accuracy not compared with some of the related latest work (e.g PD-FGC, Wav2Lip), texture quality metrics and emotion accuracy (MEAD) do not demonstrate clear improvement over state-of-the-art. The improvement in  fps is clearly compromising image fidelity metrics, especially on MEAD dataset.
•	An ablation study is needed to justify whether use of the supervised loss on 3DMM expression coefficients degrades the lip sync accuracy, otherwise the second contribution (lines 93-94) is not properly justified.

•	Unclear phrases :
o	Line 75 - introduce a 3DMM as a differentiable bridge
o	Line 97 - to prevent feature interference
•	Lack of clarity in Fig 1 block b.1 : Difficult to understand the process flow from inputs f_TEX and f_STY to the Generator network
•	Complex phrase : Line 138 - autonomously distilling nuanced emotional cues directly from the input audio’s acoustic properties.
•	“primary cause of low fidelity in talking head generation is the ambiguity of direct audio-to-pixel mapping” - this statement should be rephrased to indicate lip sync quality as fidelity might imply visual quality, which is unrelated to audio.

**Questions:**

Please see weaknesses


•	EVP extracts emotional information from audio, it does not use emotion labels as input (unlike MEAD). According to the definition of Expressive Naturalness in lines 36-37, Why is EVP marked as not possessing expressive naturalness in Table 1.
•	Please elaborate how (Lines 186-187) 3DMM parametric representation inherently disentangles the strong audio-lip correlation from weaker correlations with head pose or blinks.
•	Why does the 3DMM "differentiable bridge" perform better than supervision on 3DMM coefficients ?
•	How does the lip sync accuracy metric compare with Wav2Lip [1] , PD-FGC[4] and PC-AVS [3] ?
•	It is not clear which module introduces such a significant improvement in FPS over EAMM, EAT, SadTalker which are not diffusion-based.

---

> ### Author Response · Authors · 2025-11-21
> **Response to Reviewer Th4P Part 1.**
>
> _1. Qualitative comparisons with emotional talking head generation methods._
>
> Thank you for your insightful feedback. Following your suggestion, **we have added a new paragraph in Appendix E under "Comparison with Emotional Generation Methods"** dedicated to a direct comparison with emotional talking head generation methods.
>
> The newly added section is as follows:
>
> > **Comparison with Emotional Generation Methods.** To further validate our claims, we directly compare RETA against emotion-driven methods like EAT and EMMN in Fig. 8. These methods face two critical limitations that RETA overcomes. First, regarding expressiveness, their reliance on discrete labels results in static, "canned" emotions. In stark contrast, our completely label-free approach learns expressions directly from audio prosody. Second, in terms of real-time performance, they sacrifice speed for control. EAT (12.23 FPS) and EAMM (16.87 FPS) are far too slow for interactive use. RETA, however, delivers superior emotional fidelity at over 55 FPS (a 3.6x speedup). Thus, RETA is unique in resolving the core trade-off between expressiveness and real-time performance that has constrained prior work.
> >
>
>
>
> _2. Comparison with lip-sync focuse methods._
>
> To address this concern directly, we have **added a new section to the Appendix (e.g., Appendix E) titled "Comparison with Lip-sync Focused Methods."** This section benchmarks RETA against the very methods the reviewer mentioned (Wav2Lip and PC-AVS). The results of this new quantitative analysis are presented below:
>
> > **Comparison with Lip-sync Focused Methods**. A direct comparison with methods hyper-focused on lip-synchronization, such as Wav2Lip and PC-AVS, reveals the critical trade-off that RETA successfully resolves. These specialist models achieve top-tier Syncc scores by narrowly optimizing for audio-to-lip mapping, but this singular focus comes at a steep cost. By ignoring prosodic cues, they produce emotionally inert animations where motion is confined to the mouth region, resulting in extremely poor expressiveness (ACCe < 15%) and realism (FID > 21), as shown in Table 6. RETA, in contrast, avoids this severe compromise. It achieves a Syncc score (6.21 on HDTF) that is highly competitive and closely approaches the ground-truth level (6.49), while simultaneously delivering SOTA expressiveness (62.35% ACCe) and photorealism (14.80 FID). This superior balance is a direct result of our architectural design: the differentiable 3DMM bridge provides robust geometric guidance for precise lip articulation, while the parallel emotion pathway animates the entire face with nuanced expression. Thus, RETA proves that world-class expressiveness does not require sacrificing lip-sync quality, offering a holistic and far more practical solution.
> >
>
> | Method | Inference | MEAD PSNR ↑ | MEAD SSIM ↑ | MEAD FID ↓ | MEAD Sync_c ↑ | MEAD ACC_e ↑ | HDTF PSNR ↑ | HDTF SSIM ↑ | HDTF FID ↓ | HDTF Sync_c ↑ |
> | --- | --- | --- | --- | --- | --- | --- | --- | --- | --- | --- |
> | Wav2Lip | 29.34 | 20.72 | 0.65 | 41.27 | 6.92 | 14.26 | 21.81 | 0.63 | 21.85 | 7.31 |
> | PC-AVS | 46.24 | 16.42 | 0.47 | 46.56 | 6.21 | 13.23 | 22.14 | 0.69 | 27.33 | 6.86 |
> | RETA (ours) | 55.32 | 22.19 | 0.77 | 17.09 | 5.84 | 62.35 | 22.82 | 0.75 | 14.80 | 6.21 |
>
> _3. Regarding the Length and Scope of Comparison Videos._
>
> We have addressed these points by updating the comprehensive video demonstrations. We **have clarified this in the revised "Additional Experimental Results" section (Lines 1175-1186, Page 22)**.
>
> The revised sections are as follows:
>
> > ...Furthermore, to provide a comprehensive qualitative assessment, we present extensive side-by-side video comparisons using the HDTF and VFHQ datasets on our project page: [https://github.com/AnonymousRETA/VisualizationRETA](https://github.com/AnonymousRETA/VisualizationRETA).  These videos feature **complete sentences and paragraphs**, enabling a thorough evaluation of temporal consistency and expressive dynamics. These videos demonstrate how RETA's technical innovations translate to visual superiority. Our label-free emotion learning yields far more expressive results than lip-sync-focused methods like **Wav2Lip**, where animation is largely confined to the mouth region. Our end-to-end "differentiable bridge" avoids the error accumulation seen in GAN-based models (SadTalker, RealPortrait), producing sharper and more stable animations. Finally, our single-pass GAN generator achieves a quality competitive with diffusion models like KDTalker but at real-time speeds (>55 FPS), overcoming their critical latency issues. Thus, the qualitative results confirm RETA's ability to resolve the key trade-offs limiting prior work.
> >

---

> > ### Author Response · Authors · 2025-11-21
> > **Response to Reviewer Th4P Part 2.**
> >
> > _4. More citations._
> >
> > We have thoroughly revised our **Related Works (Section 2)**. We confirm that **PC-AVS and SyncTalk were already cited**, and we have now added discussions for all the other mentioned papers. Specifically,
> >
> > + **Page 2, Lines 107-108:**
> >
> > > More recently, SyncTalk (Peng et al., 2024) leverages a NeRF-based pipeline with a 3D facial blendshape model to enhance expression control and pose stability.
> > >
> >
> > + **Page 3, Lines 113-126:**
> >
> > > Emotional control has been a central challenge, with methods evolving from explicit supervision to more sophisticated, audio-driven techniques. Early approaches, such as the GCN-based method of Sinha et al. (2022), relied on discrete, one-hot emotion vectors, which limited expressiveness to predefined categories. A second line of work achieved more dynamic control by using external guidance, either by transferring style from a reference video, as in EAMM (Ji et al., 2022) and DREAM-Talk (Zhang et al., 2023a), or by using text prompts, as seen in Style2Talker (Tan et al., 2024b) and EAT (Gan et al., 2023). While flexible, these methods all depend on auxiliary, non-audio inputs. A more advanced paradigm aims to derive emotion directly from the rich cues within the audio signal itself, typically by disentangling it from other motion factors. PC-AVS (Zhou et al., 2021), for example, focuses on separating head pose from lip motion. More directly targeting emotion, EVP (Ji et al., 2021) uses unimodal audio cross-reconstruction, GC-AVT (Liang et al., 2022) employs input-space masking of facial regions, and PD-FGC (Wang et al., 2023) utilizes statistical decorrelation. Although these audio-driven strategies are self-contained, they often define emotion indirectly through structural or statistical separation.
> > >
> >
> > + **Page 3, Lines 136-138:**
> >
> > > More recently, EmotiveTalk (Wang et al., 2025) proposed a video diffusion framework that uses a Vision-guided Audio Information Decoupling (V-AID) approach to generate separate representations for lips and expression.
> > >
> >
> > _5. The contribution is further weakened by the fact that the generator (G) and image encoder (E_I) are borrowed directly from EDTalk, including their pre-trained weights._
> >
> > We thank the reviewer for this crucial point, which allows us to clarify the core philosophy of our work. We agree that this is an important point to make explicit, and we have **revised the introductory paragraph of our Method section (Section 3) to state our design philosophy upfront.** We respectfully argue that our use of pre-trained models is a deliberate and strategic choice that strengthens our contribution by allowing us to focus on the true bottlenecks in talking head generation.
> >
> > Our approach deliberately embraces the paradigm of **leveraging knowledge from large-scale pre-trained models** to solve complex downstream tasks. In addition to the EDTalk generator and image encoder, our framework strategically orchestrates knowledge from several powerful pre-trained models:
> >
> > + An **Audio Encoder** (**Wav2Vec 2.0**) for rich feature extraction.
> > + A **Visual "Teacher"** (**ViT**) for our cross-modal emotion distillation.
> > + A **Lip-Sync Expert** (**SyncNet**) for robust audio-visual supervision.
> >
> > This strategy offers significant advantages. It allows us to focus our research efforts squarely on our primary innovations—the novel control signals ($f_M$, $f_E$) and their integration via Hierarchical Injection—rather than on re-engineering a renderer. Furthermore, our core mechanisms are themselves _enabled_ by this approach; for instance, our cross-modal distillation would be impractical without a powerful pre-trained ViT teacher. Finally, using a fixed, SOTA generator provides a rigorous and fair ablation, demonstrating that the significant improvements in lip-sync and expressiveness are directly attributable to our unique control framework, which makes the source of our contribution scientifically unambiguous. In summary, our contribution is not a new renderer, but a novel and highly effective **control framework** that intelligently orchestrates pre-trained components to solve the generative trilemma.

---

> > > ### Author Response · Authors · 2025-11-21
> > > **Response to Reviewer Th4P Part 3.**
> > >
> > > _6. The core idea of disentangling the audio signal into geometric motion and emotion (Lines 14, 179) is not novel. This has been a central theme in many prior works, including EVP, EAMM, and PD-FGC. The paper fails to articulate what makes its disentanglement strategy a significant advancement over these existing methods._
> > >
> > > We thank the reviewer for their insightful comment, which highlights the need to clearly differentiate our work from prior disentanglement strategies. We agree that the high-level concept has been explored, and we have now added a dedicated section, "**Appendix B. Relation to Prior Disentanglement Methods**", as follows and referred the readers to this appendix in Section 2.
> > >
> > > > While prior works such as EVP (Ji et al., 2021), EAMM (Ji et al., 2022), and PD-FGC (Wang et al., 2022) also explore the disentanglement of geometric motion and emotion, RETA introduces a fundamentally different set of mechanisms that offer distinct advantages in practicality and performance. We compare our strategy against their core disentanglement techniques below.
> > > >
> > >
> > > > **Comparison with EVP**. EVP (Ji et al., 2021) operates within a unimodal framework, using an audio-to-audio cross-reconstruction objective on DTW-aligned pseudo-pairs to disentangle emotion and content entirely within the audio domain. RETA’s approach is fundamentally cross-modal. By distilling knowledge from a pre-trained visual expert, we provide a richer, visually-grounded supervisory signal that is more direct than EVP's audio-only self-reconstruction.
> > > >
> > >
> > > > **Comparison with EAMM**. EAMM (Ji et al., 2022) relies on an external emotion source video, learning to represent emotion as transferable linear displacements on unsupervised keypoints. This makes it dependent on an additional, user-provided input. In contrast, RETA is purely audio-driven, learning a dynamic emotion embedding $f_E$ directly from the audio's prosody. Our label-free, cross-modal distillation creates a self-contained system that removes this dependency.
> > > >
> > >
> > > > **Comparison with PD-FGC.** PD-FGC (Wang et al., 2022) uses statistical decorrelation to enforce feature-level independence, implicitly defining emotion as the residual information left after separating other motion factors. RETA, by contrast, employs a more direct learning signal through cross-modal knowledge distillation. We explicitly map audio prosody to a semantically rich visual emotion space, providing a stronger learning objective than relying on statistical independence.
> > > >
> > >
> > > > In summary, RETA’s disentanglement strategy is a significant advancement due to its unique combination of: (1) purely audio-driven emotion synthesis, eliminating the need for external references (vs. EAMM); (2) a cross-modal, visually-grounded learning paradigm that provides a richer supervisory signal than unimodal or statistical methods (vs. EVP, PD-FGC); and (3) the integration of this advanced emotion representation with a novel, end-to-end differentiable 3DMM bridge for superior lip-sync. This holistic design enables RETA to uniquely resolve the generative trilemma.
> > > >
> > >
> > >
> > > _7. It is not clear which specific module or design choice leads to such a significant FPS improvement over other non-diffusion-based methods._
> > >
> > > To provide a complete and transparent picture of our model's efficiency, we have **added a new section to Appendix F in our revised manuscript**, which includes an in-depth discussion of the results.
> > >
> > > The newly added Appendix is as follows:
> > >
> > > > **Comparison to non-diffusion-based methods:** Compared to methods employing decoupled or cascaded pipelines, such as SadTalker and EAMM, RETA's unified architecture offers significant speed advantages. These prior methods break the synthesis process into multiple discrete stages. For instance, SadTalker is bottlenecked by slow inter-process communication via disk files, while EAMM employs a complex cascade of separate modules for motion superposition and CPU-based filtering. RETA, by contrast, integrates all steps into a single, end-to-end GPU-native framework, eliminating these systemic bottlenecks. When compared to other models like EAT, their performance limitations arise from different factors. EAT relies on a computationally heavy Transformer architecture to predict its intermediate 3D keypoint representation. While this avoids a multi-stage pipeline, the complexity of the Transformer itself becomes the primary computational bottleneck. In contrast, RETA leverages a lightweight, differentiable 3DMM prior, which allows us to use a much more efficient convolutional generator. This design choice enables robust geometric guidance without the computational overhead of large-scale attention mechanisms, and the generator is further optimized for efficient batch processing.
> > > >

---

> ### Author Response · Authors · 2025-11-21
> **Response to Reviewer Th4P Part 4.**
>
> _8. Clarification on paper presentation._
>
> + _Unclear Phrases: The following phrases require clarification: "to prevent feature interference" (Line 97) and "autonomously distilling nuanced emotional cues directly from the input audio’s acoustic properties" (Line 138)_
> + _Confusing Figure: The process flow in Figure 1 (b.1) is difficult to follow. Please clarify how inputs f_TEX and f_STY are processed and used by the Generator network._
> + _Imprecise Wording: The statement "primary cause of low fidelity in talking head generation is the ambiguity of direct audio-to-pixel mapping" should be rephrased. The term "fidelity" is ambiguous; the context suggests the issue is primarily with lip-sync quality, not overall visual quality._
>
>
>
> **1. On Unclear Phrases:**
>
> + **Regarding "to prevent feature interference" (Line 97):** We agree this phrase needs more context. "Feature interference" refers to the conflict that arises when coarse-grained structural signals (like geometric motion, $f_M$) and fine-grained detail signals (like subtle emotion, $f_E$) are naively combined and fed into all layers of the generator. Our generator operates on a coarse-to-fine principle: early layers establish the basic facial structure and pose, while later layers add nuanced details. Our **Hierarchical Injection** strategy aligns with this by injecting the geometric prior $f_M$ into the early layers and the emotion embedding $f_E$ into the later ones. This separation ensures that the signals for large-scale motion do not disrupt the synthesis of fine-grained emotional details, and vice versa. To address this concern, **we have revised the sentence to be more explicit:**
>
> > …to prevent interference between the coarse-grained geometric signals (for structure) and the fine-grained emotion signals (for detail).
> >
>
> + **Regarding "autonomously distilling nuanced emotional cues..." (Line 127):** This phrase describes our label-free emotion learning process. "Autonomously" means the process requires no manual emotion labels. "Distilling from... acoustic properties" refers to learning from the raw audio waveform's characteristics (e.g., prosody, pitch, cadence), not just semantic content. We will clarify this as:
>
> > ..autonomously learns to extract nuanced emotional cues from the input audio’s acoustic properties. This is achieved via cross-modal distillation from a visual expert, which transfers knowledge without requiring explicit emotion labels.
> >
>
> **2. On the Confusing Figure (Figure 1 (b.1)):**
>
> We appreciate the feedback on Figure 1's clarity. The diagram aims to show how the identity features, $f_{TEX}$ and $f_{STY}$, are integrated. They follow two distinct pathways within the generator, which is more clearly detailed in the architecture diagram (Figure 4) in Appendix C.2.
>
> + **$f_{TEX}$ (Spatial Texture Features):** This is a pyramid of _spatial_ feature maps from the image encoder's intermediate layers. They retain high-fidelity texture and local details. As shown in Figure 4(b), they are fed via **skip connections** to the corresponding resolution layers in the generator. They act as the detailed "content canvas" that is warped and animated.
> + **$f_{STY}$ (Global Style Code):** This is a single, _non-spatial_ vector from the final layer of the image encoder. It captures the holistic appearance and identity. It is injected into the generator's layers via **AdaIN** (Adaptive Instance Normalization), where it acts as a global "style" anchor to ensure identity consistency throughout the animation.
>
> We have revised the caption for Figure 1 to **add a cross-reference to Figure 4** for a more detailed architectural view.
>
> **3. On Imprecise Wording ("fidelity"):**
>
> The core issue of "direct audio-to-pixel mapping" is not just about visual quality, but the inherent **one-to-many ambiguity** of the mapping itself. A single phoneme can be spoken with various head poses, blinks, and micro-expressions, making it difficult for a model to predict the correct motion. This ambiguity primarily affects **motion correctness and lip-sync accuracy**, not just photorealism. We have rephrased the sentence for better precision:
>
> > As highlighted by Zhou et al. (2020), a primary challenge in talking head generation is the ambiguity of direct audio-to-pixel mapping, which can lead to inaccurate lip movements and unnatural motion.
> >

---

> ### Author Response · Authors · 2025-11-21
> **Response to Reviewer Th4P Part 5.**
>
> _9. Clarification on main contribution of differentiable bridge._
>
> We agree that empirically justifying our second contribution is crucial. As the reviewer's advice, we have **expanded the "3D Mesh Alignment and Quality" section in our manuscript** to provide the necessary justification. The newly added text in the manuscript now explicitly clarifies our rationale and provides empirical proof:
>
> > ...Forcing a model to match pre-computed, and potentially noisy, ground-truth coefficients locks in suboptimal geometric predictions early in the pipeline. These errors then propagate and are amplified during the final rendering stage. Our "differentiable bridge" approach, in contrast, allows the model to learn a geometric prior that is optimally suited for the final image synthesis task, rather than merely mimicking an intermediate representation. The end-to-end backpropagation from the final image loss guides the model to discover the most effective geometric cues for generating a realistic output. Furthermore, to empirically demonstrate that our end-to-end approach still learns a geometrically sound representation, we evaluated the Lip Vertex Error (LVE) of our generated meshes against a SOTA 3D-based model (UniTalker~\citealp{fan2024unitalker}) as a proxy for ground-truth geometry. Our model achieves a strong LVE of 4.30, confirming that even without explicit supervision, our differentiable bridge effectively learns a high-fidelity geometric prior.
> >
>
> _10. Clarification on the classification of EVP._
>
> We thank the reviewer for this insightful comment and for highlighting a potential ambiguity in our categorization of EVP in Table 1. The reviewer is correct that EVP extracts emotional cues directly from audio during **inference** and does not require an explicit emotion label as input from the user. Our original rationale for categorizing it as "Label-Dependent" stemmed **from its training process.** To improve clarity and avoid any misinterpretation, we have followed the reviewer's suggestion and **remove EVP from the "Label-Dependent Emotional Models" category in Table 1.**
>
> _11. Please elaborate how (Lines 186-187) 3DMM parametric representation inherently disentangles the strong audio-lip correlation from weaker correlations with head pose or blinks._
>
> This point is already explained in detail in **Appendix B.1** in the original version of this paper.
>
> Indeed, the "inherent disentanglement" we refer to stems from the fundamental design of a 3D Morphable Model (3DMM). A 3DMM is not a monolithic representation; it is a statistical model constructed with separate, near-orthogonal parameter sets that control distinct aspects of facial variation. Specifically, as detailed in Appendix C.1, a face mesh is synthesized using parameters for:
>
> + **Expression** ($p^{exp}$): A vector controlling non-rigid deformations like lip movements, smiles, and frowns.
> + **Pose** ($r$, $t$): Rotation and translation vectors controlling rigid head movements.
> + **Identity** ($p^{id}$): A vector controlling the static, underlying bone structure and facial shape.
>
> This structured parameterization transforms the ill-posed, one-to-many problem of mapping audio directly to pixels into a set of more constrained, well-defined regression tasks. The disentanglement occurs because different components of the audio signal have vastly different correlational strengths with these separate parameter sets:
>
> + **Strong Correlation (Audio ****↔**** Lip Motion)**: The phonetic content of speech (visemes) has a strong, direct, and predictable correlation with the subset of $p^{exp}$ parameters that control the lips and jaw. For example, the phoneme /b/ consistently maps to a specific closed-lip shape. Our model learns to map audio features to these specific expression parameters, ensuring robust lip-sync.
> + **Weaker Correlation (Audio ****↔**** Head Pose)**: The prosody, rhythm, and cadence of speech may correlate with head pose (e.g., nodding for emphasis), but this relationship is much weaker, more stochastic, and highly person-dependent. It does not follow a one-to-one mapping.
> + **No Correlation (Audio ****↔**** Blinks)**: Eye blinks are physiological reflexes that are almost entirely uncorrelated with the content of the audio signal.

---

> ### Author Response · Authors · 2025-11-27
> **Response to Reviewer Th4P**
>
> Dear Reviewer Th4P,
>
> We hope our follow-up response and supplementary experiments help to address your questions. Your review is very valuable, and we would appreciate your further feedback. Of course, please let us know if this or any other point remains unclear, as we would be happy to elaborate further.
>
> Thank you for your valuable time!
>
> Sincerely,
>
> Authors of Paper 5010

---

> > ### Comment · Reviewer_Th4P · 2025-11-27
> > **Post rebuttle**
> >
> > GIven the rebuttal I am increasing the score marginally as I still notice lip sync problem for the word "those","republicans"  etc in the generated videos. Lot of changes needs to be in the paper too.

---

> > > ### Author Response · Authors · 2025-11-28
> > > **Response to Reviewer Th4P**
> > >
> > > Thank you very much for re-evaluating our work and for the score increase. We are especially grateful for your continued engagement and your specific feedback, which we have found very helpful.
> > >
> > > **1. On the Manuscript Changes:**
> > > Regarding your comment that "a lot of changes needs to be in the paper too," we wanted to respectfully clarify that **we have already performed a comprehensive revision of the manuscript** based on the valuable feedback from all reviewers. The revised paper submitted with our rebuttal already incorporates these substantial changes. To make this clear, here is a summary of the major updates we made:
> > >
> > > + **New Zero-Shot Experiments (Appendix G, Table 8):** Added a new evaluation on the challenging VFHQ dataset to demonstrate generalization.
> > > + **New Computational Analysis (Appendix F, Table 7):** Provided a detailed breakdown of inference speed, latency, and VRAM usage.
> > > + **New Empirical Justification (Section 3.3):** Added a quantitative Lip Vertex Error evaluation to validate our "differentiable bridge."
> > > + **New Appendix on Disentanglement (Appendix B):** Differentiated our method from prior works like EVP, EAMM, and PD-FGC.
> > > + **Extensive Revisions for Clarity:** Thoroughly revised the Related Works, Method, and Limitations sections based on reviewer feedback.
> > >
> > > We hope this confirmation assures you that we have diligently addressed the required modifications within the paper itself.
> > >
> > > **2. On the Zero-Shot Video Results:**
> > > We appreciate you pointing out the subtle lip-sync imperfections on specific words. Your observation is correct. It is important to contextualize this result: the video in question is from our new, highly demanding **zero-shot experiment**. This test uses the "in-the-wild" VFHQ dataset, where **both the subject's identity and the driving audio were completely unseen during training**.
> > >
> > > + **Difficulty of the Task:** Achieving perfect accuracy on every single viseme in such an unconstrained, out-of-distribution scenario remains a significant open challenge for the field. We believe that our model's ability to maintain overall stability and produce largely intelligible speech on a long, unseen clip underscores its strong generalization and robustness.
> > > + **Additional Demonstrations:** To further showcase our model's capabilities on more typical inputs, we have added several more video examples to our project page. Specifically, we demonstrate results for a source image from the **unseen VFHQ dataset** when driven by two distinct audio types: one from another completely unseen dataset (a fully unseen scenario) and another from the HDTF dataset (testing an unseen identity with a more familiar audio domain). We invite you to view these new demos, which exhibit the high-fidelity lip-sync that RETA consistently achieves, without the specific issues you noted: [https://github.com/AnonymousRETA/VisualizationRETA](https://github.com/AnonymousRETA/VisualizationRETA)
> > > + **Objective Quantitative Metrics:** Finally, while individual video examples are crucial, we believe the **quantitative metrics** presented in our tables provide a more objective and holistic measure of performance. Metrics like **Sync_c**, which are averaged over the entire test set, reflect the model's overall performance rather than its behavior on a few challenging words. Our strong quantitative results across all benchmarks (e.g., Sync_c of **6.21** on HDTF and **5.63** on VFHQ) confirm the overall high quality and accuracy of our method.
> > >
> > > Thank you once again for your constructive feedback. We hope this detailed clarification on the completed manuscript changes and the context of our video results is helpful for your final evaluation.

---

### Author Response · Authors · 2025-11-23
**General Response**

We sincerely thank all the reviewers for their constructive feedback and valuable suggestions, which have significantly contributed to improving the quality of our manuscript. We have prepared a comprehensive reply to address every point raised. For clarity, the reviewer's comments are presented in italics, followed by our response. Quotations from the revised paper are included in blockquotes to highlight our changes. Unless otherwise specified, all references to pages, equations, and sections relate to the original submission.

Specifically, the major revisions and additions are summarized below:

1. **Zero-Shot Generalization Experiments (Appendix G, Table 8, Figure 9)**: To address concerns about generalization to "in-the-wild" scenarios (Reviewers 12nH, iuLE), we conducted a new set of zero-shot experiments on the challenging **VFHQ dataset**. The strong quantitative results, presented in a new Table 8, demonstrate RETA's robustness and superior performance on completely **unseen subjects and complex backgrounds**.
2. **Detailed Computational Cost Analysis (Appendix F, Table 7)**: To provide a transparent and in-depth view of our model's efficiency (Reviewers iuLE, Th4P), we added a new Appendix F with a **comprehensive computational analysis**. This includes a new Table 7, which benchmarks RETA's inference speed (FPS), latency, and VRAM usage against key baselines, quantitatively confirming our model's real-time capabilities (>55 FPS).
3. **Empirical Justification for "Differentiable Bridge" (Section 3.3)**: To empirically justify the effectiveness of our differentiable 3DMM bridge (Reviewer Th4P), we conducted **a new evaluation of the Lip Vertex Error (LVE)** against a SOTA 3D-based model. This quantitative result has been added to the manuscript, proving that our end-to-end approach learns a geometrically sound representation without explicit supervision.
4. **Clarification on Novelty and Disentanglement (Appendix B):** To better contextualize our contributions and differentiate our work from prior methods (Reviewer Th4P), we added **a new Appendix B ('Relation to Prior Disentanglement Methods')**. This section provides a detailed, head-to-head comparison of our disentanglement strategy against prior works like EVP, EAMM, and PD-FGC.
5. **Comprehensive Video Demonstrations**: In response to feedback on the scope of our qualitative results (Reviewers T1UX, Th4P), we have updated our project page with longer, **more comprehensive video comparisons on the HDTF and VFHQ datasets**. These videos feature complete sentences and paragraphs to better showcase temporal consistency and expressive dynamics.
6. **Extensive Manuscript Revisions for Clarity**: We have thoroughly **revised and expanded multiple sections** of the manuscript for clarity based on reviewer feedback. This includes:
    - Adding more citations and discussions in **Related Works** (Reviewer Th4P).
    - Clarifying our design philosophy regarding the strategic use of pre-trained models in the **Method section** (Reviewer Th4P).
    - **Refining the explanation** of our Hierarchical Injection mechanism and its distinction from prior work (Reviewer T1UX).
    - Expanding the **Limitations and Future Work** section to explicitly address aspects like the absence of eye blinking and the dependency on the visual teacher (Reviewers T1UX, 12nH).

These comprehensive updates, including new experiments, detailed analyses, and extensive clarifications, directly address the reviewers' concerns and substantially strengthen our paper. We believe the revised manuscript now more clearly demonstrates RETA's effectiveness in resolving the key trade-offs in talking head generation.

---

### Meta-Review · Area_Chair_3CXn · 2025-12-30

**Summary:**

This paper focuses on real-time animation generation of expressive talking heads. It proposes an end-to-end generation framework to address lip-syncing and dynamic facial expression control, ultimately achieving 55fps dynamic generation. The paper received two positive and two negative scores, and the negative comments remained after discussion. This has made the paper controversial, and the area chair carefully reviewed these comments, which are summarized below.

**Reviewer Concerns:**

+ Reviewer Th4P's main concern was a qualitative comparison with EAT, EMMN, EVP, and MEAD. The reviewer provided a detailed discussion of the paper's main innovations and contributions compared to existing work. The authors detailed some experimental and interpretive revisions in their response. However, the reviewer still felt there were issues with the video's effects and that the paper required significant revisions, giving a relatively negative evaluation. Although the authors explained the reasons for these problems and the difficulty of the task, the area chair acknowledged the reviewer's concerns.

+ Reviewer 12nH's main concern was the complexity of the method, including control over facial expressions and types, and the lack of rigorous experimental and quantitative evaluation. The authors provided detailed responses, which were acknowledged by the area chair.

+ Reviewer T1UX expressed concerns about the anonymous supplemental submission date, as well as the main experimental results, video length, effectiveness, and the representation of blinking, a crucial element for digital humans. The authors acknowledged the submission delay, provided updated videos, and engaged in multiple rounds of discussion. However, T1UX still expressed dissatisfaction with the video quality and acknowledged existing shortcomings. The area chair accepted T1UX's assessment and recognized significant room for improvement.

+ Reviewer iuLE primarily worried about the impact of the static teacher model on performance and the relatively limited dataset used by the authors. The authors acknowledged these shortcomings and will clarify the limitations. Furthermore, they provided additional baseline model experiments. These responses, to some extent, address the reviewers' concerns.

**Reviewer Scores:**

In summary, despite multiple rounds of discussion, some reviewers still expressed significant concerns, particularly regarding the effectiveness in real-world cases and the results demonstrated in the video demo. This is crucial for visual generation tasks. After discussion, the paper still received two clear negative comments (4 points), and even after the authors uploaded new video materials, the reviewers' questions could not be fully addressed. The area chair carefully read these comments and the revised paper, and believed that the paper still had significant room for improvement.

---

### Decision · Program_Chairs · 2026-01-26

Reject